# Normative growth trajectories of fetal brain regions validated by satisfactory maturation of neurodevelopmental domains at 2 years of age

Madeleine K. Wyburd [1,26] ✉, Stephen H. Kennedy [2,26], Michelle Fernandes [2,3,4], Nicola K. Dinsdale[1], Linde S. Hesse[1], Robert B. Gunier [5], Leila Cheikh Ismail[6], Eric O. Ohuma [7], Michael G. Gravett[8], Manorama Purwar[9], Wu Qingqing[10], Adele Winsey[2,4], Enrico Bertino[11], Yasmin Jaffer[12], Maria Carvalho [13], Fernando C. Barros[14], Alan Stein [15,16,17], Alison J. Noble [18], Zoltán Molnár [19], Mark Jenkinson [20,21,22], Thomas E. Nichols [20,23], Stephen Smith[20], Zulfiqar A. Bhutta [24,25], Aris T. Papageorghiou [2,4], Jose Villar [2,4,27] & Ana I. L. Namburete [1,20,27] ✉

We previously constructed a qualitative, 3D ultrasound derived atlas of the normative spatiotemporal dynamics of fetal brain maturation. Here, using the same healthy multi-national cohort, we applied deep learning methods to 4205 fetal brain scans from 18–27 weeks' gestation, to produce an extensive, quantitative description of the growth of 16 fetal brain structures associated with satisfactory domain-specific neurodevelopmental scores at 2 years of age. The methodology, which is publicly available, takes less than 10 seconds per scan. We define 28 region-specific, functionally relevant, normative growth trajectories, a ratio between the relative volumes of the insular (rILV) and parietal (rPLV) lobes reflecting asynchronous maturation of fetal brain regions, and introduce a fetal brain maturation index that quantifies biological age and deviations from chronological age. Finally, the very low percentage of variance explained by between site differences (0.6% to 5.8% of the total variance) reinforces a fundamental biological principle: fetal growth and development across populations with diverse ancestries is similar provided that environmental constraints on growth are minimal.

The use of imaging tools to describe the normative early maturation of human brain structures is essential for monitoring differences over time and linking structural organisation to developmental functions in later life. To date, the most comprehensive description of human brain growth was compiled from over 100 studies that collected magnetic resonance imaging (MRI) data between 115 days post-conception and 100 years of age[1].

Unfortunately, only three studies contributed fetal data to those brain growth charts.

All are limited by their cross-sectional design, small sample size during fetal life, methodological heterogeneity, and lack of postnatal follow-up which means the normative neurodevelopment of the fetal sample cannot be confirmed. In addition, although MRI scans have been performed more than once during pregnancy to produce fetal

brain volumetric trajectories[2], healthy women with low-risk pregnancies rarely have repeated MRI scans and certainly not in large numbers. Including only healthy subjects in studies to define normative growth and development (i.e., a prescriptive approach) is essential, as recommended by the World Health Organization (WHO)[3]. However, none of the recent brain charts across the human lifespan appear to have included pregnant women from low-risk populations who meet WHO's strict criteria[1,4].

Using the 3D US scans obtained from over 4000 healthy pregnancies in the international, population-based INTERGROWTH-21st Fetal Growth Longitudinal Study (FGLS), and by following WHO recommendations[3], we recently constructed a qualitative digital atlas of the normative spatiotemporal dynamics of fetal brain maturation[5]. We now report, from the same dataset, a quantitative description of the growth of specific brain regions with corresponding normative ranges. To do so at scale, intracranial structures need to be labelled manually, which is extremely time consuming[6]. Although deep learning (DL) approaches have been proposed to automate structure labelling[7,8] they do not always work on fetal US images, as large acoustic shadows obstruct key anatomical regions, leading to poor quality segmentations[6].

To avoid these limitations, we apply a DL method that guides structural labelling in regions with acoustic shadows to produce anatomical segmentations of fetal brains[9,10]. From the standardised FGLS 3D US dataset[5], we labelled 11 brain structures and five cortical regions and extracted 28 image-derived phenotypes (IDPs) that were also used to produce a fetal brain maturation index, which quantitatively describes normative development. We focused on the gestational period between 18 and 27 weeks as we have previously shown that fetal cranial growth trajectories diverge within the 20-25-week gestational age window predicting differential growth and neurodevelopmental outcomes at 2 years of age[11].

## Results

### Image-derived phenotypes of the fetal brain

A total of 4321 women, enrolled in FGLS during the first trimester of pregnancy, had live singleton births in the absence of severe maternal conditions or congenital abnormalities detected by US or at birth, all of whom had very accurate estimates of gestational age. (Fig. 1a). Thereafter, 765 (17.7%) of these fetuses were excluded from the analysis due to missing postnatal follow-up data ($n = 681$) or evidence of severe morbidity ($n = 84$). Of the remaining 3556 fetuses, 2906 had at least one 3D US scan between 18 and 27 weeks, the gestational period of interest.

To validate the postnatal development at 2 years of age of the normative cohort, we studied a sub-sample of 1112 children (38.3% of the eligible fetuses) using the INTERGROWTH-21st Neurodevelopmental Assessment (INTER-NDA) tool[12] and Cardiff Visual Acuity and Contrast Sensitivity tests for binocular vision[13].

As we have measured small size differences within different regions of the fetal brain, we adopted a very conservative approach to the analysis of the neurodevelopmental and vision assessments. Thus, we excluded the 3D US scans of the 101 eligible fetuses that scored in the bottom 3% on any one of the INTER-NDA domains or had low scores for visual acuity (< 24 at 50 cm for 6/m) or sensitivity (< 33.3) at 2 years of age.

Hence, to construct normative growth charts, we included a total sample of 2805 fetuses (prospectively monitored from early pregnancy onwards) that had 4205 3D US scans in the 9-week period, which was the focus of the present study. Sensitivity analysis confirmed the appropriateness of combining the IDPs of the fetuses with and without a developmental assessment at 2 years of age (Supplementary Table 1 and Supplementary Fig. 1). Finally, the sub-sample ($n = 1011/2906$) and the proportion of the total FGLS population ($n = 1181/4321$) evaluated at 2 years of age were similar on the full set of variables measured (Supplementary Table 2).

The median interval between scanning sessions was 5 weeks (mean 4.61; SD 0.58) as per protocol, and 49.6% ($n = 1390$) of the women had at least two scans (median 1.0; mean 1.50; SD 0.5; range 1–3 weeks) (Fig. 1b, Supplementary Table 3). Using two DL convolutional networks[9,10], each of the 4205 3D US scans was labelled into 11 structures. From each scan, the total brain volume (TBV) and cavum septum (CSP) was extracted across both hemispheres. However, as the proximal hemisphere is frequently obscured by acoustic shadowing from the skull, only the distal hemisphere (i.e., the side furthest from the US probe) was labelled into the cortical plate (CoP), white matter (WM), deep grey matter (DGM), cerebellum (CB), thalamus (Th), lateral posterior ventricle horns (LV), choroid plexus (ChP), frontal horns (FH) and brainstem (BS), as shown in Fig. 2. Each structure's volume was then computed.

As the Sylvian fissure (SF) is the epicentre of change in the second trimester of pregnancy, an additional measure of SF depth (SFD) was included. We further characterised the complex expansion and folding of the CoP, a process driven by neuron migration and differentiation[14], by measuring four CoP components: volume (CoPV), surface area (CoPSA), thickness (CoPT) and depth (CoPD). CoPV, CoPT and CoPD were extended into five functionally delimited lobes: frontal (FL),

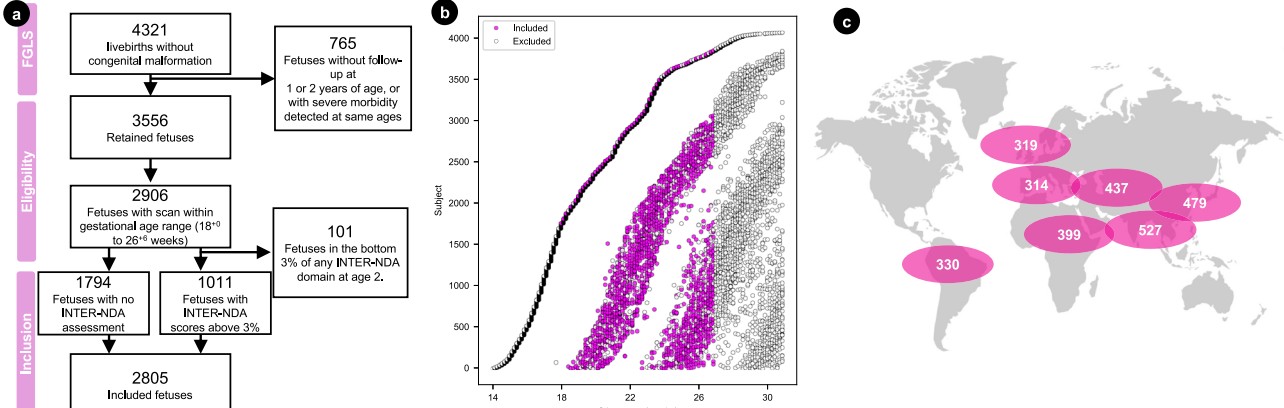

**Fig. 1 | Summary statistics of fetuses from the INTERGROWTH-21st Fetal Growth Longitudinal Study (FGLS) included in the present study. a** Flowchart summarising the inclusion criteria, number of fetuses, and number of 3D ultrasound scans remaining at each step. **b** Lasagna plot of the number of scanning sessions (visits) per fetus in the FGLS dataset. Of the 2906 fetuses included in the analyses of brain regions between 18 and 27 weeks' gestation, 49.5% ($n = 1433$) had at least two scans. **c** Map showing the contribution of each study site to the total dataset, designed by Freepik (http://www.freepik/).

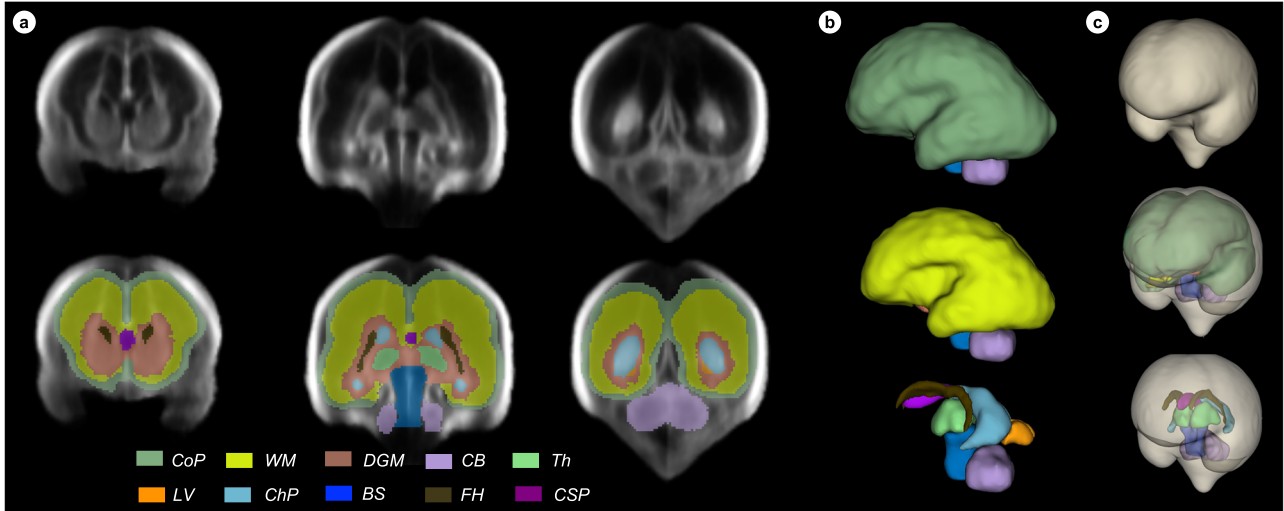

**Fig. 2 | Proposed tissue parcellation.** Proposed tissue parcellation protocol on the fetal brain ultrasound atlas at 22 weeks' gestation (Namburete, 2023), shown in (**a**) the coronal view, (**b**) in 3D and (**c**) in 3D and encompassed by the total brain (TB). The structures labeled are cortical plate (CoP), white matter (WM), deep grey matter (DGM), cerebellum (CB), thalamus (Th), lateral posterior ventricle horns (LV), choroid plexus (ChP), frontal horns (FH), brainstem (BS) and cavum septum (CSP). A full description of the image-derived phenotypes is shown in Table 1. It should be noted that, although both the left and right hemispheres were labelled on each atlas, only the distal hemisphere was labelled on individual scans, with the exception of TB and CSP.

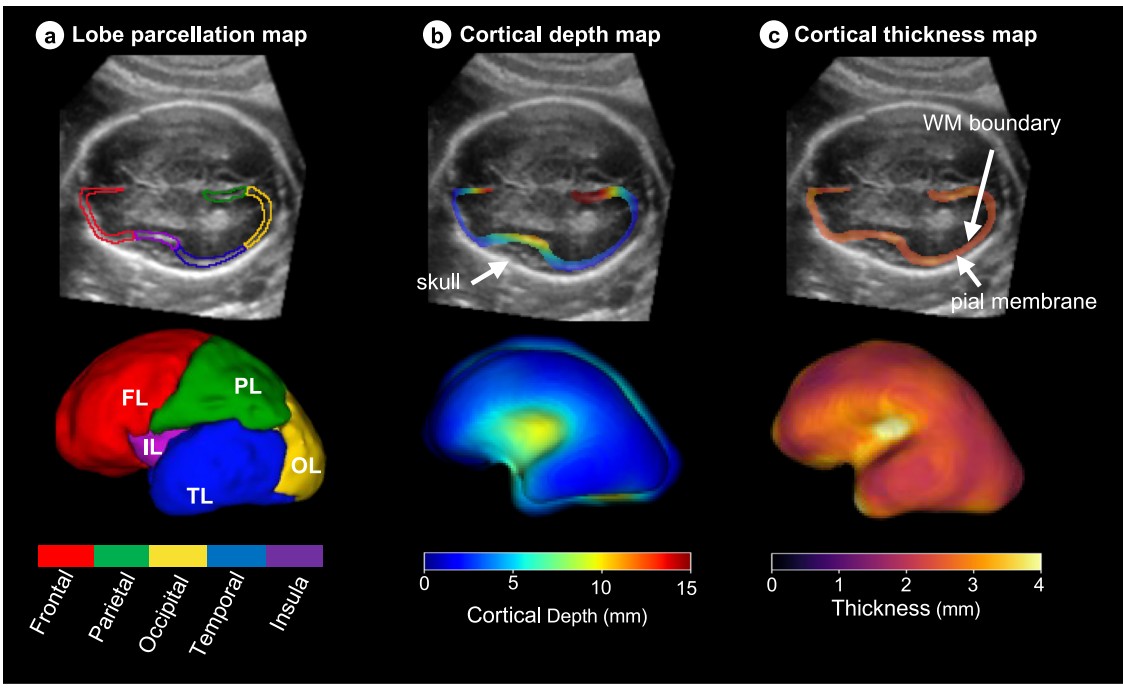

**Fig. 3 | Measures of the fetal brain cortex. a** The cortical lobes' parcellation for an individual fetal brain shown overlaid on the scan (top row) and as a 3D reconstruction (bottom row). The same is displayed for (**b**) cortical depth and (**c**) cortical thickness. The cortical depth measures the distance from the skull to each position within the cortex. The 3D projection shows the white matter cortical boundary. Cortical thickness measures the shortest distance between the pial membrane and white matter boundary, the 3D projection is on the pial membrane. These maps provide complementary visualisation of fetal cortical maturation, integrating morphological and microstructural developmental trajectories. The cortical parcellation map are for: frontal lobe (FL), temporal lobe (TL), parietal lobe (PL), occipital lobe (OL) and insular lobe (IL). A full description of the image-derived phenotypes is shown in Table 1.

temporal (TL), parietal (PL), occipital (OL) and insular (IL) (Fig. 3), enabling average regional measures of each lobe.

Thus, in total, 28 IDPs were automatically extracted measuring volume, surface area, cortical depth and thickness from the 4205 3D US scans of 2805 fetuses, taking a total of 15 h on a single Nvidia A10 GPU (Nvidia Corporation, Santa Clara, CA, USA). Table 1 presents an overview of the IDPs studied and their acronyms. Except for TBV and CSPV, the measurement of each structure was possible only on the distal hemisphere, as the shape of the skull creates an acoustic cavity that often obstructs the proximal hemisphere with strong shadows and thus, too much structural information is lost to guide the segmentation algorithm.

**Agreement with magnetic resonance imaging studies**

Our algorithms labelled structures based on the brain features manually annotated on an US atlas[5] by a researcher (M.K.W)

**Table 1 | An overview of the acronyms describing the 28 image derived phenotypes (IDPs)**

| 11 regions of interest | | | |
|---|---|---|---|
| Both | TB | Total Brain | |
| | CSP | Cavum septum | |
| Distal hemisphere only | CoP | Cortical plate | |
| | WM | White matter | |
| | DGM | Deep grey matter | |
| | CB | Cerebellum | |
| | ChP | Choroid plexus | |
| | LV | Lateral posterior ventricle horns | |
| | FH | Frontal horns | |
| | BS | Brainstem | |
| | Th | Thalamus | |
| **5 cortical parcellations** | | | |
| Distal hemisphere only | **FL** | Frontal lobe | |
| | **PL** | Parietal lobe | |
| | **OL** | Occipital lobe | |
| | **TL** | Temporal lobe | |
| | **IL** | Insular lobe | |
| **Measurement type** | | | |
| | **V** | Volume | |
| | **SA** | Surface area | |
| | **D** | Depth | |
| | **T** | Thickness | |
| | **SFD** | Sylvian Fissure depth | |

The volume of each region and parcellation was assessed, with the mean thickness and depth also measured for the cortical parcellations. Additional measures included cortical plate (CoP) SA and Sylvian fissure depth. TB and CSP was labelled across the entire brain, whereas, the remaining regions were only segmented on the distal hemisphere. Full list of the 28 IDPs: Total brain volume (TBV), cortical plate volume (CoPV), white matter volume (WMV), deep grey matter volume (DGMV), cerebellum volume (CBV), thalamus volume (ThV), lateral posterior ventricle horns volume (LVV), choroid plexus volume (ChPV), frontal horns volume (FHV), brainstem volume (BSV), cavum septum volume (CSPV), cortical plate surface area (CoPSA), Sylvian fissure depth (SFD), frontal lobe volume (FLV), temporal lobe volume (TLV), parietal lobe volume (PLV), occipital lobe volume (OLV), insular lobe volume (ILV), frontal lobe depth (FLD), temporal lobe depth (TLD), parietal lobe depth (PLD), occipital lobe depth (OLD), insular lobe depth (ILD), frontal lobe thickness (FLT), temporal lobe thickness (TLT), parietal lobe thickness (PLT), occipital lobe thickness (OLT) and insular lobe thickness (ILT).

experienced in the field of brain imaging, and then verified by A.I.L.N, the senior investigator. The structures were selected based on overlap with existing MRI atlases[8,15] and those clearly visible in our digital atlas[5].

To ensure the structures were accurately delineated, the volumes of the atlas labels were compared to those from five, publicly available, labelled fetal MRI atlases[8,15–19], as shown in Fig. 4. The volumes of the CoP, CB, Th, WM and CSP corresponded strongly to the previous MRI labels, validating the accuracy of these tissue segmentations. A large difference was seen for the BS volume, which is expected as the US field of view is much higher in the BS (Fig. 4). Differences are also seen in the ventricular system (VS) and DGM. In the US atlas, there is a clear separation between the FH, LV and ChP[5], whereas, in MRI studies these structures are usually grouped[8,15–18]. The VS volume varied across the MRI atlases due to different definitions, with some including the third ventricle and others only segmenting the ChP[18]. In the US atlas, the VS only includes the FH, LV and ChP, missing the inferior horn: hence, a lower volume compared to MRI measures is expected, as seen in Fig. 4c. The DGM shows a similar pattern, with inconsistent labelling or grouping between MRI atlases, leading to large uncertainty in DGM volume across the methods.

## Brain structural similarities across study sites

The contribution of fetuses ($n = 2805$) from each site was 11.2% Turin (Italy), 11.4% Oxford (UK), 11.8% Pelotas (Brazil), 14.2% Nairobi (Kenya), 15.6% Muscat (Oman), 17.1% Beijing (China) and 18.8% Nagpur (India). A full breakdown of the subject and scan contribution from each study site is shown in Supplementary Table 3 and Fig. 1c.

To ensure the data could be pooled to enable normative growth trajectories to be constructed, we conducted sensitivity analyses to assess the effect of removing a single site's data, one at a time for each of the 28 IDPs. There were no substantive effects on the remaining pooled sample's $3^{rd}$, $50^{th}$ and $97^{th}$ centiles for any brain structure. Figure 5a shows a representative example of the sensitivity analyses derived for TBV by excluding, one at a time, the samples from China, India, Kenya and the UK, whose general populations are popularly believed to differ in anthropometric sizes[20]. Excluding data from each of these four different populations had minimal impact on the centile estimates derived from the remaining pooled sample. Excluding the other countries similarly had little effect on the centiles (data not shown).

Supplementary Table 4 presents the percentage of the total variance attributed to differences amongst study sites (between sites differences) estimated using variance component analysis. Amongst the 28 IDPs, the proportion of the total variance adjusted by sex and gestational age at assessment explained by study site differences was only 1.6% for TBV; for the other intracranial structures, site differences explained less than 5.8% of the total variance for each brain region (Fig. 5b). Consistent with this minimal variance explained, site effects were not statistically significant in the mixed-effects models once sex and gestational age were included.

Finally, we calculated the standardised site difference (SSD), which allows for direct comparisons of volumetric brain measures across study sites, standardised by the corresponding pooled SD and adjusted for gestational age. SSD was computed for each of the seven sites and each IDP at three 3-week time-windows: $18^{+0}$ to $20^{+6}$, $21^{+0}$ to $23^{+6}$, and $24^{+0}$ to $26^{+6}$ weeks' gestation, to account for longitudinal growth. Out of the possible 588 SSD calculations (Fig. 5c; Supplementary Tables 5), 532 (90.5%) were within -0.5 and 0.5 units of SD, the interval prespecified in the FGLS protocol as adequate for combining data from all sites based on the cut-off point used in the WHO Multicentre Growth Reference Study (MGRS) that produced the prescriptive WHO Child Growth Standards[21]. For TBV, three out of 21 comparisons were outside the -0.5 to 0.5 units of SD; however, these can be explained by the skewed mean average gestational age in each sample within the 3-week time-window, shown in Supplementary Table 5. For instance, the mean gestational age in the Beijing dataset was 18.5 weeks in the $18^{+0}$ to $20^{+6}$ weeks' gestation window, a week less than the value for the sites combined (19.5 weeks). Therefore, we would expect the volume for this study site to be smaller than the sites combined. The results of these analyses clearly show that the seven study populations were sufficiently similar for the data to be pooled.

## Normative fetal brain growth

Scatter plots of observed growth data for each IDP, and the fitted $3^{rd}$, $50^{th}$ and $97^{th}$ smoothed centiles according to gestational age (in weeks) are shown in Fig. 6 and Fig. 7. To produce these normative equations the IDPs were pooled across fetal sex, cerebral hemisphere, and whether or not neurodevelopment was assessed at 2 years of age. Out of the possible 504 SSD calculations (Supplementary Table 1,6,7), 503 (99.8%) were within -0.5 and 0.5 units of SD. We excluded any measures that were more than 4 SD of all sites' gestational age-specific mean. Any such outlier values were excluded from subsequent analyses to ensure that spurious measures did not bias the growth models. The number of included scans per IDP is shown in Supplementary Table 8, alongside the corresponding equations for the mean and SD from the fractional polynomial regression

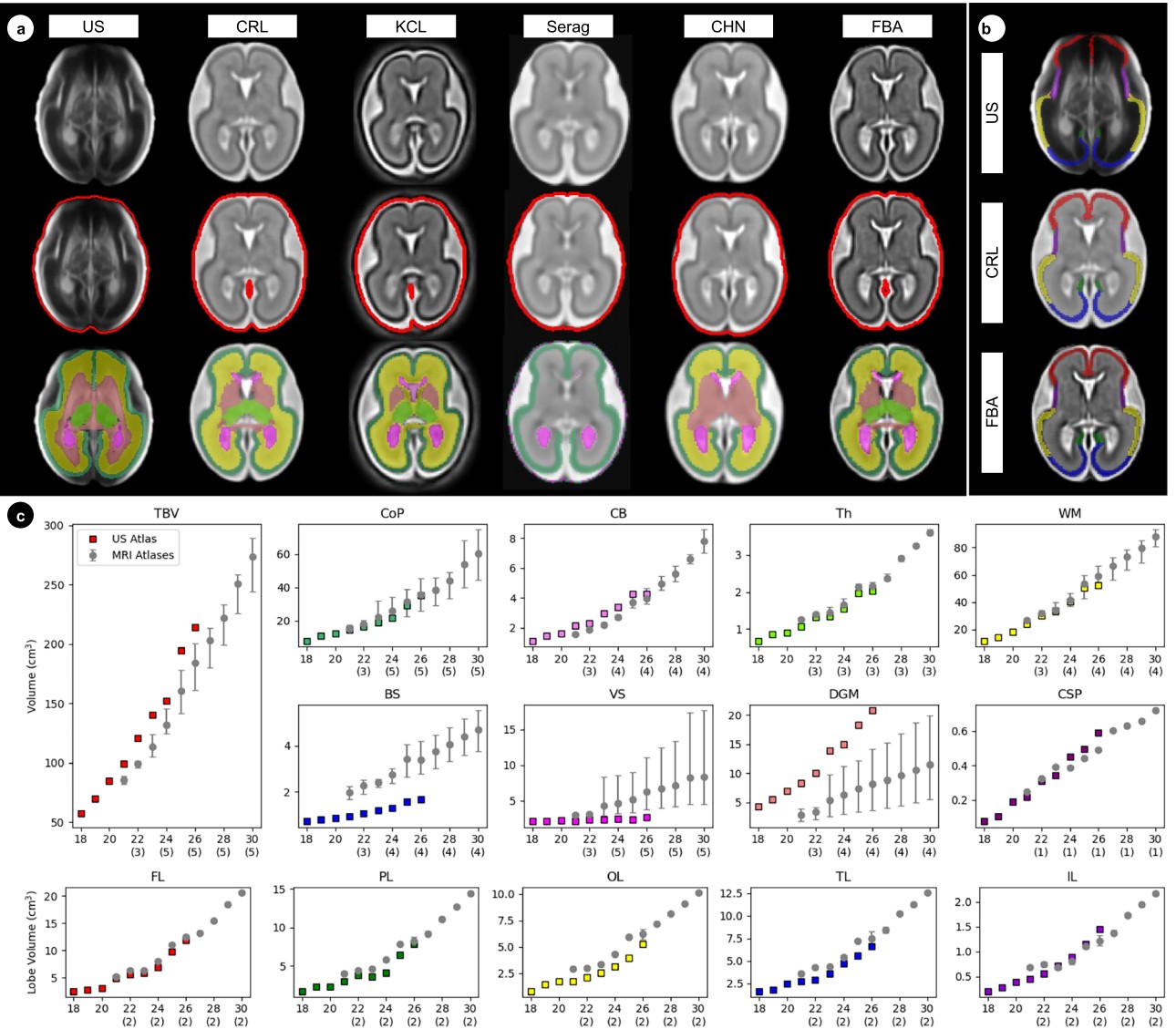

**Fig. 4 | Comparison between the atlas labels on the ultrasound (US) and five magnetic resonance imaging (MRI) atlases. a** The correspondence between MRI-atlas labels and our US-atlas tissue labels and (**b**) lobar parcellations at 24 weeks' gestation, using frontal (red), parietal (green), occipital (yellow), insula (purple) and temporal (blue) lobes. **c** The volume measures of each atlas-labelled structure across both hemispheres for each atlas. Each atlas spans a one-week period, e.g., the 26[th]-week atlas spans 26[+0days] to 26[+6days]. The MRI labels show the mean (o) with the error bars showing the standard deviation across the datasets. The number in brackets under each week of gestation shows how many MRI atlases were used for that structure. For instance, three MRI atlases had deep grey matter labelled at 22 weeks' gestation. The MRI atlases included are: CRL (Gholipour et al.[15]), KCL (Uus et al.[19]), Serag (Serag et al. [18]), CHN (Xu et al.[16]) and FBA (Wu et al.[17]).

models for each IDP, allowing readers to calculate any centiles or z-scores according to gestational age (in weeks). Table 2 shows the equation for TBV; the remaining 27 equations are shown in Supplementary Table 8.

Gestational week-specific comparisons between empirical centiles and observed values showed excellent agreement (Supplementary Fig. 2 and Supplementary Fig. 3). Overall, there was increasing, non-linear growth over the 9-week study period and the mean differences between smoothed and observed centiles for the 3[rd], 50[th] and 97[th] centiles were small for all brain structures.

The centiles reveal the trajectories of absolute volumetric growth of the brain structures, which all increased with gestational age. However, when the volume IDPs were normalised by TBV, as shown in Supplementary Fig. 4, the relative size of most structures decreased with advancing gestational age. Notably, the relative volume of the ChP decreased ~three-fold.

### Insular to parietal lobe volume ratio, each relative to the total cortical plate volume

The relative growth of each CoP lobe, normalised by total CoPV, is shown in Supplementary Fig. 5. The relative volume of the IL (rILV) increased with gestational age (Fig. 8a), consistent with the opercularisation of the Sylvian fissure from mid-gestation[22–24]. In contrast, the proportional decrease in relative PL volume (rPLV) likely reflects region-specific, temporally hierarchical maturational processes, including dendritic remodeling and synaptic pruning[25–27]. Although most synaptic pruning occurs postnatally[28], histological and MRI studies indicate this process, particularly within the transient subplate zone, commences prenatally and displays marked regional heterogeneity[25,29].

The subplate and intermediate zones (often collectively segmented as 'white matter' in MRI analyses) exhibit distinct developmental trajectories, reflecting regional differences in neuronal and axonal dynamics that likely underpin the observed

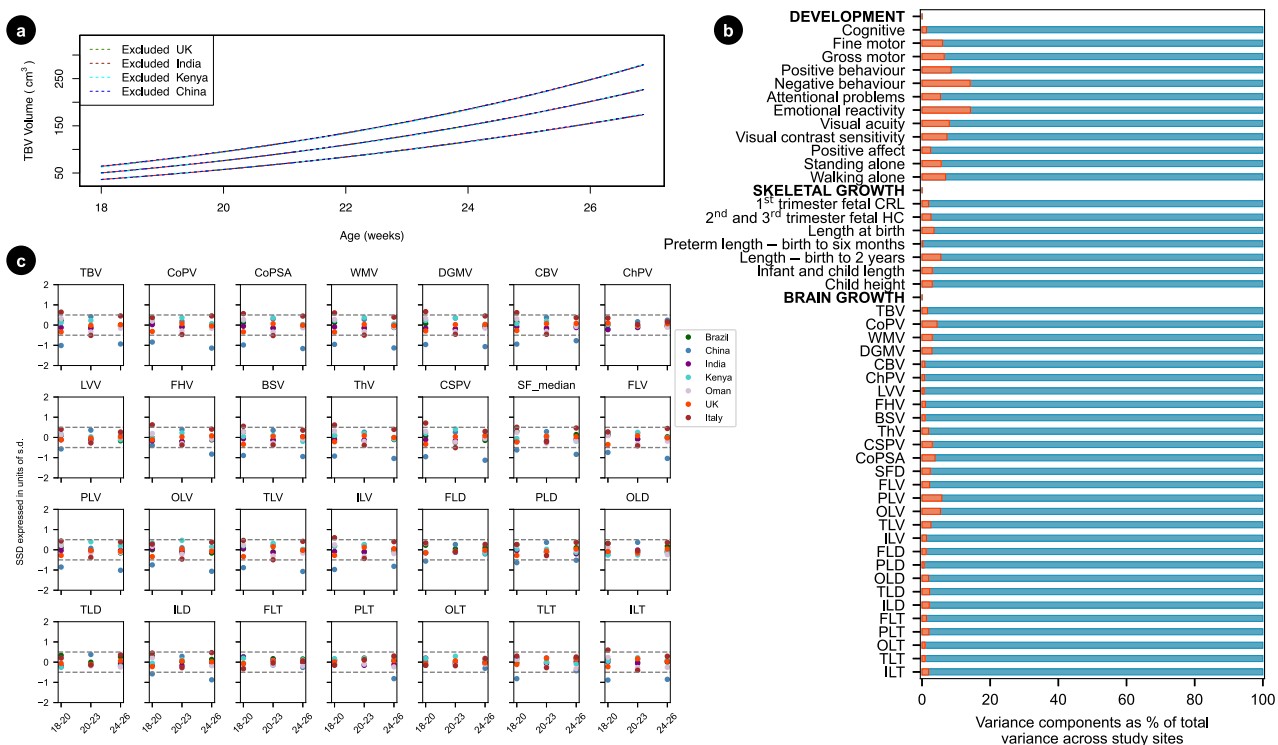

**Fig. 5 | The effect of study site on fetal brain trajectories. a** Re-generating the fetal trajectories excluding one site at a time (**b**) cross-site variance (**c**) standardised site differences SSD, all values given in Supplementary Table 5. All 28 image derived phenotypes IDPs are included: total brain volume (TBV), cortical plate volume (CoPV), white matter volume (WMV), deep grey matter volume (DGMV), cere-bellum volume (CBV), thalamus volume (ThV), lateral posterior ventricle horns volume (LVV), choroid plexus volume (ChPV), frontal horns volume (FHV), brainstem volume (BSV), cavum septum volume (CSPV), cortical plate surface area (CoPSA), Sylvian fissure depth (SFD), frontal lobe volume (FLV), temporal lobe volume (TLV), parietal lobe volume (PLV), occipital lobe volume (OLV), insular lobe volume (ILV), frontal lobe depth (FLD), temporal lobe depth (TLD), parietal lobe depth (PLD), occipital lobe depth (OLD), insular lobe depth (ILD), frontal lobe thickness (FLT), temporal lobe thickness (TLT), parietal lobe thickness (PLT), occipital lobe thickness (OLT) and insular lobe thickness (ILT).

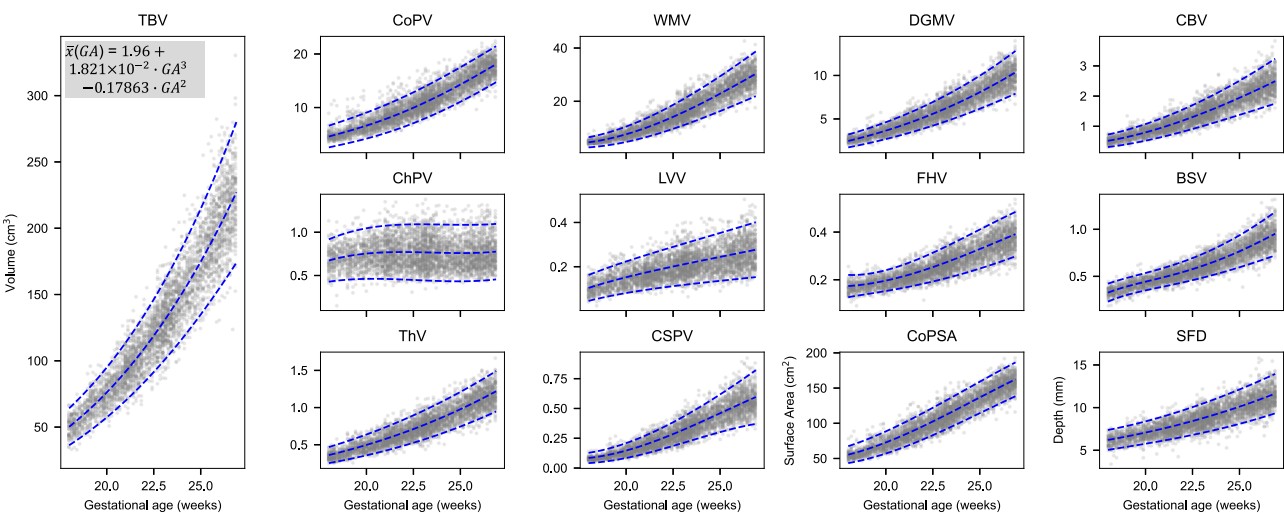

**Fig. 6 | Volumetric growth trajectories with fitted percentiles lines at the 3rd, 50th and 97th centiles.** Growth charts showing total brain volume (TBV), cortical plate volume (CoPV), white matter volume (WMV), deep grey matter volume (DGMV), cerebellum volume (CBV), thalamus volume (ThV), lateral posterior ven-tricle horns volume (LVV), choroid plexus volume (ChPV), frontal horns volume (FHV), brainstem volume (BSV), cavum septum volume (CSPV), cortical plate sur-face area (CoPSA) and Sylvian fissure depth (SFD). The equation for the mean is written on TBV, for the remaining equations see Supplementary Table 8. All IDPs except TBV and CSPV, measure only the distal hemisphere. The number of scans used to create each trajectory is given in Supplementary Table 8.

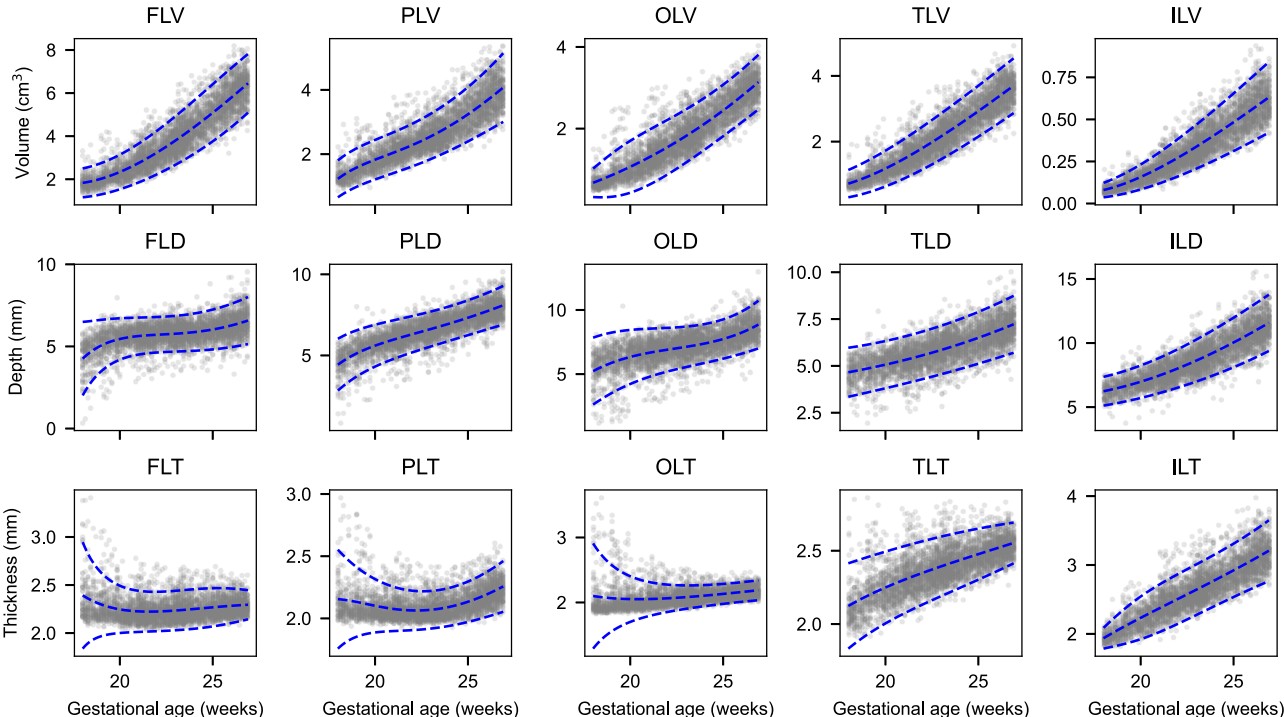

**Fig. 7 | Cortical growth trajectories with fitted percentiles lines at the 3rd, 50th and 97th centiles.** These trajectories show frontal lobe volume (FLV), parietal lobe volume (PLV), occipital lobe volume (OLV), temporal lobe volume (TLV) and insular lobe volume (ILV), frontal lobe depth (FLD), parietal lobe depth (PLD), occipital lobe depth (OLD), temporal lobe depth (TLD), and insular lobe depth (ILD), frontal lobe thickness (FLT), parietal lobe thickness (PLT), occipital lobe thickness (OLT), temporal lobe thickness (TLT) and insular lobe thickness (ILT). Each of these IDPs is measured from the distal hemisphere. The number of scans used to create each trajectory is given in Supplementary Table 8.

### Table 2 | Formulae for total brain volume

|        | Number included scans |      | Regression Equation |
|--------|-----------------------|------|---------------------|
| **TBV** | 4196 | Mean | $1.954510 + 0.018205 \times GA^3 + -0.178633 \times GA^2$ |
|        |      | SD   | $-0.702623 + 0.150265 \times GA$ |

Equations for estimating the mean and SD for total brain volume (TBV); the remaining 27 equations are in Suppl Table 8. For SD, the reported equation is on the log scale and must be exponentiated to obtain the standard deviation.

relative volumetric changes, particularly in association areas such as the PL[25].

As the rPLV progressively declined between 18 and 27 weeks' gestation, the rILV steadily increased, resulting in a concomitant rise in the rILV:rPLV ratio. We have estimated normative values for rILV:rPLV with the corresponding centiles according to gestational age (Fig. 8b). Thus, this ratio provides a snapshot of the cortical maturation of the fetal brain and captures the asynchrony in growth between these two cortical lobes.

**Fetal brain maturation index**

Comparing biological and chronological ages is commonly used in the adult MRI literature to determine if cohorts or individuals are ageing at the expected rate[30], and differences between the two are associated with disease and cognitive decline[30–32].

Here, we compare fetal brain maturation (as a marker of biological age) with our very accurate estimate of gestational (chronological) age by training a random forest to assess each fetus' gestational age at each scan using the 28 IDPs. The method assumes that brain development is consistent across healthy populations, as shown above.

Three additional factors (fetal sex, study site and distal hemisphere) were included as confounding factors in the model. Consistent with our previous findings[5], their contribution to the total variance was minimal (0.50%, 1.06% and 1.25%, respectively), and was therefore regressed out of the data.

The full set of 28 deconfounded features was used as the input for the random forest model, which accurately predicted gestational age with a mean absolute error of only 3.9 days in relation to fetal chronological age. The result was similar to the 3-day accuracy previously achieved at 20–30 weeks' gestation using the 2D US FGLS data and a machine learning approach[33]. Hence, in this dataset of healthy accurately dated fetuses, we demonstrate that biological and chronological age align very closely (R = 0.954; ICC = 0.955), enabling us to produce a fetal brain maturation index based on regional brain structure growth.

The six most informative features that contribute to the model across the gestational age range studied: TBV, CoPSA, WMV, DGMV, ILV and CBV (Supplementary Fig. 6). Including only these IDPs in the model led to an agreement between the two predictions of 4.2 days (R = 0.95), indicating that although these are the most informative features, the remaining 22 IDPs still provide a useful indication of maturation.

The contribution of each of the 28 IDPs at the three individual time-points (18, 22 and 26 weeks' gestation) is shown in Supplementary Fig. 6.

## Discussion

Using DL methods, we have presented here a detailed, quantitative description of the normative growth and maturation of 16 key fetal brain structures during the second trimester of pregnancy; a critical

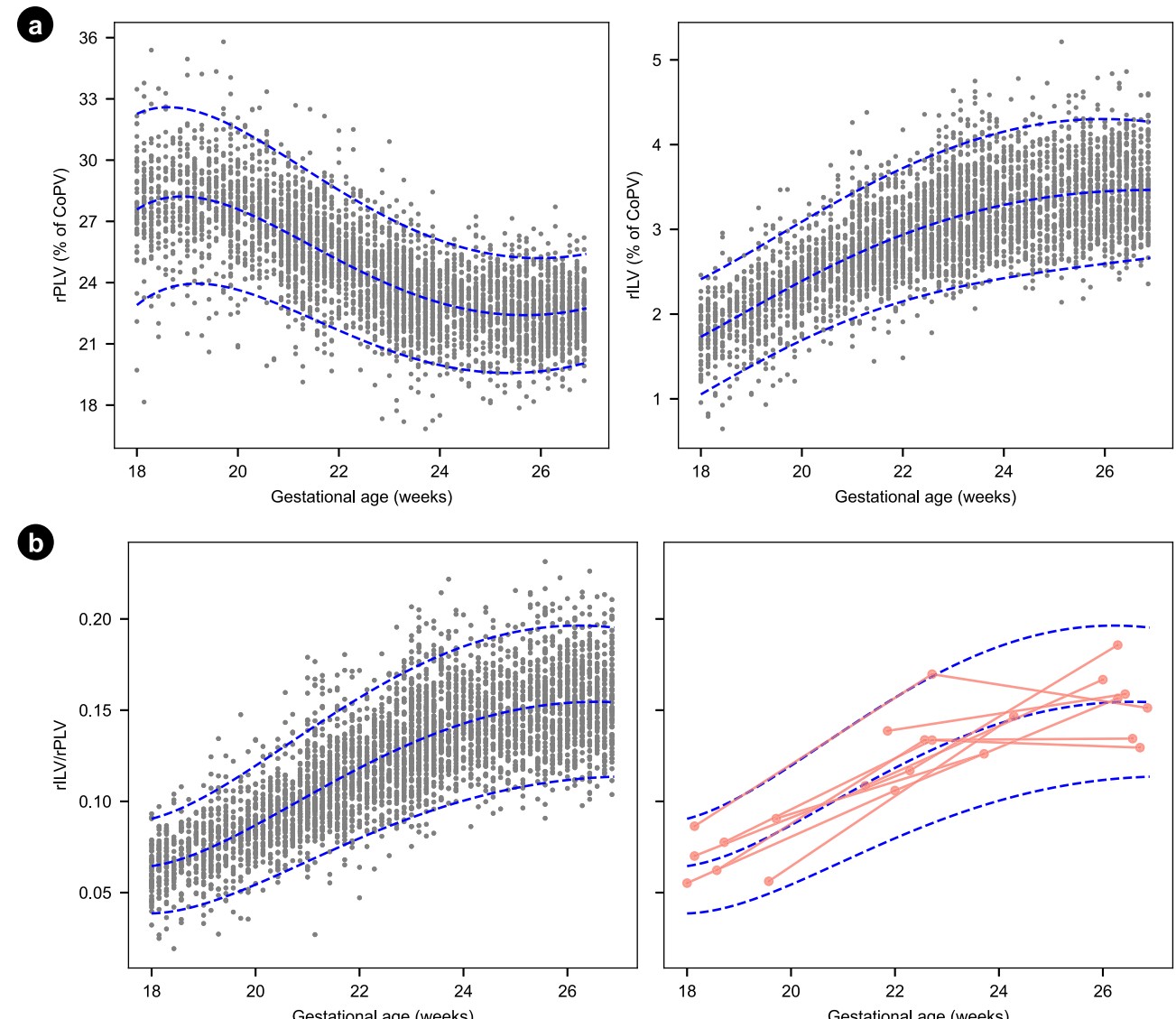

**Fig. 8 | Relative cortical lobe growth and ratio. a** Relative parietal lobe volume (rPLV) and relative insular lobe volume (rILV) from the distal hemisphere. Both volumes are relative to total cortical plate volume (CoPV). Left panel: progressive decline in rPLV with advancing gestational age, consistent with the relative volumetric redistribution of the developing CoP. Right panel: increase in rILV reflecting the accelerated maturation of the insular region. **b** Insular-Parietal lobes Volume Ratio (rILV/rPLV) from 18 to 27 weeks' gestation. The rILV/rPLV ratio shows the evolving temporal asynchrony between these cortical regions. Longitudinal growth of 10 randomly subjects is shown in pink. The blue dash lines show the 3[rd], 50[th] and 97[th] centiles, with individual values in the background. Relative growth for the remaining lobes is shown in Supp Fig. 2.

window during which faltering growth of the whole cranium is associated with delayed neurodevelopment at 2 years of age[11].

We also describe a ratio (rILV:rPLV) representative of the temporally asynchronous, relative volumetric growth of the IL and PV lobes between 18 and 27 weeks' gestation. IL US measures in fetuses with early onset growth restriction are reduced compared to appropriately grown fetuses[34], and reduced ILV (measured with MRI) in late onset small-for-gestational-age fetuses is associated with poorer neurobehavioural outcomes in early neonatal life[35]. Hence, given the interconnected functional roles of these two cortical lobes, we speculate that deviation from the normative values of the rILV:rPLV ratio, because of an adverse intrauterine environment, may lead to differences in sensory integration, self-awareness and higher-order cognition in early child development.

In addition, we have produced a fetal brain maturation index that is a learned, weighted summary of all 28 IDPs based upon the features most informative of healthy maturation. We plan to use the index in the future to quantify the degree of departure from chronological age across the participants in the INTERBIO-21st Fetal Study[11] who have different risk profiles at this key stage of brain development.

The inclusion of CoPSA and ILV in the list of informative features is interesting because cortical folding is an important neurodevelopmental process that yields increased CoPSA and ILV, allowing more neurons to be arranged within a limited skull volume, which is associated with increased cognitive abilities[14,36–38]. The emergence and accelerated growth of some secondary fissures, especially SF operculisation, occurs in the second trimester[39], the period of study here, and might therefore explain why CoPSA and ILV are so informative of gestational age.

Importantly, the children studied at 2 years of age had satisfactory growth and neurodevelopment particularly in domains involved in language acquisition and comprehension that are associated with FL and TL function. These healthy children, whose intrauterine brain growth was measured during FGLS, were evaluated at 2 years of age as

performance then is predictive of intelligence, school attainment, adult nutrition and human capital in high-, middle- and low-income settings[40–44].

Reference charts describing volumetric growth of fetal intracranial structures using MR[45,46] and US[47,48] imaging have been published previously, but from single centres with much smaller sample sizes ($n = 68$–344) and only one study included neurodevelopmental assessment at 2 years of age[45]. Interestingly, those authors reported that, in fetuses with congenital heart disease, global fetal brain volumetric growth was the most consistent predictor of the variance attributable to the known risk factors across neurodevelopmental domains at 2 years of age, accounting for 10% to 21% of the total 18% to 45%[45].

In our study, the percentage of variance explained by between-site differences ranged from 0.6% to 5.8% of the total variance. Of the 588 comparisons using SSDs, 91% were between ±0.50 units of the pooled SD for the corresponding item. Therefore, our findings complement those studies of skeletal growth from early pregnancy to 2 years of age[49], newborn size[20], infant neurodevelopment and associated behaviours[50], brain size[5] and child growth[21], which consistently demonstrate, amongst different ethnic populations in diverse regions of the world, that only a very small percentage of the total variance in these measures is explained by differences among populations (Fig. 5b) when constraints on human growth are minimal. In addition, our use of GAMLSS explicitly models non-linear developmental trajectories and variance across gestation, thereby mitigating potential site-by-age confounding effects without the need for additional harmonisation methods.

The manual atlas labels describing average normative growth corresponded well with previous MRI atlases for most structures (Fig. 4), and individual trajectories also overlapped. For example, the median CoPV range of 3 to 18 cm$^3$ reported here (Figs. 6, 7) is comparable to the 5–22 cm$^3$ range in a previous MRI study, with lobar volumes also showing high compatibility. Similar agreement was also seen for CBV and DGMV. Further, this approach was recently validated on an independent dataset of 90 fetal brain ultrasound volumes with paired, same-day MRI[51], where volume measures were highly correlated, with good to excellent agreement in the subcortical structures.

We observed a median cortical plate thickness of 2–3 mm between 18 and 27 weeks, which is consistant with 1.92–2.84 mm reported in an MRI-based study[52]. However, cortical thickness estimates are variable across studies[16,52,53]. Differences in absolute values likely reflect modality- and method-related factors: in ultrasound, cortical plate boundaries are defined by echogenicity, whereas in MRI they are delineated using tissue relaxation contrasts. Such cross-modality variation has also been described for other fetal brain structures[51]. We note, however, that cortical thickness estimates show greater variability than other measures, consistent with previous MRI reports[53], reflecting both methodological differences and the intrinsic difficulty of delineating this boundary during early development. Because cortical thickness was computed voxel-wise from the CoP label, precision is inherently limited by voxel size (±0.6 mm in this study), segmentation accuracy, and potential partial-volume effects[54]. In future work, adapting surface-based techniques to operate robustly on single-hemisphere ultrasound data may help address these limitation[55]. Importantly, the developmental trajectories across gestation are consistent with MRI findings[16], supporting the biological validity of our measures.

Importantly, some structures are more clearly distinguished in US compared to MRI[5], enabling us to extract previously unreported IDPs and describe their longitudinal growth for the first time. In US studies, the FH, LV and ChP are reported separately[5], whereas, in MRI studies, these structures are usually grouped[8,15]. Disentangling the VS into FH, LV and ChP, shows that each structure grows at a very different rate (Fig. 6), information previously lost from MRI analysis. Notably,

although ChP increases with gestational age, the growth is more gradual than for the other structures, and when normalised by TBV, it actually decreases (Supplementary Fig. 4)[56].

Our automated pipeline analysis extracted 28 IDPs from the fetal brain in less than 10 sec per 3D US scan, as compared to ~20 h when done manually. Our approach not only segments 11 structures and 5 functionally delineated cortical lobes from each scan, but also performs complex analyses, such as measures of CoPSA, SFD, CoPT, CoPD and CoPSA, which were highly informative of gestational age and could, therefore, be used clinically to estimate gestational age in the second trimester of pregnancy if a first-trimester US scan has not been performed. This potential use is only possible because of the speed achieved by the analysis pipeline, which will be made open access, enabling future research to: a) perform in depth neuroanatomical research from 3D US and b) compare individuals to the normative growth trajectories and fetal brain maturation index presented here.

The strengths of the study were: 1) The DL algorithms delineated the whole brain (Dice overlap = 0.95) and individual brain structures (Dice overlap = 0.78) accurately in less than 10 seconds per scan, overcoming the time-consuming task of manual segmentations previously required to study volumetric growth; 2) The IDPs studied were highly informative of chronological age, demonstrating their potential use as a "fetal maturation index" for the study of less healthy cohorts, provided that gestational age is accurately determined; 3) We adopted the same prescriptive approach as the WHO MGRS, which produced the WHO Child Growth Standards[21], in keeping with the recommendation of the 1995 special WHO Expert Committee[3]. Thus, we prospectively enrolled a cohort of healthy, educated and adequately nourished pregnant women whose risk of adverse maternal and perinatal outcomes (including preterm birth and fetal growth restriction) was low based on their individual clinical profiles and the socioeconomic, demographic and environmental characteristics of the underlying seven geographically diverse populations[20]; 4) We followed a sample of 36% of mother-infant eligible dyads from early pregnancy to 2 years of age using standardised measures, feeding practices and data collection methods. We also, very conservatively, excluded from the analyses 101 children that had scores, on any of the INTER-NDA domains, below the 3rd centile of the normative values. Detailed information was also obtained about the socioeconomic, education and environmental backgrounds of the families selected at both population and individual levels[57].

The principal limitation of the study were: 1) We pooled the size measures between hemispheres rather than report values for each hemisphere. However, this was justified as the standardised differences between the hemispheres were within the WHO recommended range for comparing human growth patterns (±0.5 units of SD) and confirmed by consistent growth trajectories across both hemispheres (Supplementary Fig. 7). Importantly, this strategy is supported by our previous findings showing that the asymmetries in fetal brain structures during the gestational age window studied here are driven by shape, not size[5]; 2) While the segmentation methods were thoroughly tested and verified on a representative subset of scans (2% of the total dataset; 4205 scans in total), their performance was not explicitly verified across the entire dataset. Although DL-based segmentation could in principle hallucinate anatomy in poor-quality scans, we did not observe this. Manual review of 70 representative scans and independent validation in 90 volumes with same-day MRI demonstrated anatomically faithful labels, supports the reliability of our approach[51]. Given the scale of the data and the practical constraints on manual review, this approach reflects a considered and pragmatic balance between methodological rigour and feasibility. The analysis therefore rests on the reasonable assumption that the model generalises comparably across the full dataset, supported by its strong performance on the verified subset.

In summary, we have presented evidence that, when environmental, nutritional and health care constraints on growth are minimal, functionally important areas of the human brain develop similarly across diverse populations in the second trimester of pregnancy, a key period of neurogenesis. The findings are consistent with previously reported similarities in the same cohort in neurodevelopmental outcomes at 2 years of age[50], and skeletal growth during fetal life, at birth and in early childhood[49]. The pooled IDPs enabled us to construct normative growth trajectories, a marker of asynchronous volumetric growth of the IL and PV lobes and a fetal brain maturational index (biological fetal age), which when combined with our published digital atlas[5], that shows the spatiotemporal dynamics of the maturation of the fetal brain (https://intergrowth21.com/research/brain-atlas-project), should be valuable as a tool to investigate factors that modify developmental processes and affect cognitive function in childhood[11].

Finally, our findings reinforce a fundamental biological principle: most of the observed differences in growth and neurodevelopment across populations or regions of the world are primarily due to socioeconomic, educational and class disparities, rather than genetic ancestry[58–60]. The notion that brain size and neurodevelopment are related to skin colour or 'race'[61] is simply false.

## Methods

The INTERGROWTH-21st Study and its ancillary studies were approved by the Oxfordshire Research Ethics Committee "C" (reference: 08/H0606/139), the research ethics committees of the individual participating institutions, as well as the corresponding regional health authorities where the project was implemented. All mothers provided written informed consent for the use of their clinical data.

An overview of the final dataset used here (4205 scans from 2805 subjects), including sex, age at scan and site, is provided in Supplementary Table 3.

### Study participants and fetal ultrasound scans

A detailed description of the study protocol has been published elsewhere, including the selection criteria at cluster and individual level[20,62]. In brief, the population-based Fetal Growth Longitudinal Study (FGLS) of the INTERGROWTH-21st Project was conducted between 2009 and 2016 in eight delimited, geographically diverse, urban areas: Pelotas (Brazil), Turin (Italy), Muscat (Oman), Oxford (UK), Seattle (USA), Shunyi County in Beijing (China), the central area of Nagpur (India), and the Parklands suburb of Nairobi (Kenya). However, 3D US scans from the Seattle site were not included in this analysis because of missing follow-up data.

Participating women, who initiated antenatal care before 14 weeks' gestation, were selected based on WHO criteria for optimal health, nutrition, education and socioeconomic status needed to construct international growth standards[3]. Hence, they had low-risk pregnancies that fulfilled well-defined and strict inclusion criteria at both population and individual levels[20].

The last menstrual period (LMP) was used to calculate gestational age (± 7 days) provided that: a) the date was certain; b) the woman had a regular 24-32 day menstrual cycle; c) she had not been using hormonal contraception or breastfeeding in the preceding 2 months, and d) any discrepancy between the gestational ages based on LMP and crown-rump length (CRL), measured by ultrasound at 9 + 0 to 13 + 6 weeks from the LMP was ≤7 days, using the formula described by Robinson & Fleming[63]. The CRL technique was standardised across sites and all ultrasonographers were trained uniformly[64].

3D US scans of the fetal head were acquired every 5 ± 1 weeks from 14 weeks' gestation to delivery (i.e., 14-18, 19-23, 24-28, 29-33, 34-38 and 39-42 weeks' gestation) to explore their use for undertaking 2D measurements[65], and for additional exploratory analyses[66,67].

Dedicated ultrasonographers performed the scans using identical, commercially available, equipment (Philips HD-9, Philips Ultrasound, Bothell, WA, USA), with a curvilinear abdominal 3D transducer (V7-3). The 3D US probe was positioned such that the central axial view was collected at the level of the thalami, and the angle of insonation was adjusted to include the entire skull (~70°) for a typical volume acquisition time of 4 sec. We conducted centralised hands-on training of sonographers, and the Oxford-based Ultrasound Quality Control (QC) Unit regularly carried out site-specific standardisation procedures to ensure proper use of the US equipment and protocol adherence.

FGLS has produced international growth standards for fetal head, abdomen and femur[62]. Crucially, at 2 years of age, these infants had satisfactory growth, neurodevelopmental milestones and associated behaviours[49,50], confirming the cohort's suitability for producing growth standards.

### Fetal brain analysis

**Fetal brain volume acquisition.** Post-acquisition processing of the 3D volumes followed a minimal pipeline for processing fetal brain US scans. This involved a previously published sequence of manual and automated procedures.

In brief, the 3D volumes were first visually inspected to ensure they were of good quality and free from motion artifacts and acoustic shadows that occluded the structures of interest, ultimately excluding scans from 6% (259/4321) of the participants.

The images were then resampled to an isotropic voxel size of $0.6 \times 0.6 \times 0.6$ mm$^3$ using trilinear interpolation. Across the dataset, initial voxel sizes ranged from 0.14–0.76 mm (SD 0.04) in the $x$-dimension, 0.30-1.01 mm (SD 0.15) in the $y$-dimension, and 0.27–1.91 mm (SD 0.23) in the $z$-dimension, with a median of $0.32 \times 0.51 \times 0.85$ mm$^3$. As the scans were collected at the axial view of the fetal head, the $x$-direction in the raw volumes consistently corresponds to the axial plane. This was confirmed empirically by examining the rotation parameters from the alignment step across all 4,205 scans, which showed two narrow peaks for separated by $\pi$ (or 180°), corresponding to fetuses facing left versus right. The narrowness of these peaks indicates only modest deviation from the intended axial orientation (Supplementary Fig. 8). In 3D ultrasound, the reported "slice thickness" reflects voxel spacing in the reconstructed volume, not physical slice separation as in MRI, and therefore does not imply angular misalignment.

This interpolation approach avoids pixel skipping and minimises the risk of aliasing artifacts. Given that most acquisitions had finer in-plane resolution and only modestly coarser through-plane resolution, the resampling primarily involved upsampling rather than downsampling, thereby further reducing the likelihood of aliasing. Voxel size distributions were highly uniform across imaging sites and gestational ages, indicating effective cross-site harmonisation of acquisition protocols (Supplementary Fig. 8).

All images were then intensity-normalised using per-volume min-max scaling, such that the minimum voxel intensity was set to 0 and the maximum to 1, thereby standardising all inputs while preserving relative contrast. Per-volume min-max normalisation results in minor variation in the absolute normalised intensity values of specific tissues across scans, but preserves the relative intra-volume contrast (e.g., between cortical plate and CSF). Tissue intensities were found to be relatively consistent across the dataset (see Supplementary Fig. 8), reflecting the standardised scanning protocol and the fact that ultrasound contrast arises from the physical acoustic properties of tissues. Therefore, more complex MRI-style intensity standardisation was not required; simple per-volume min−max normalisation was sufficient to standardise input scale while preserving relative contrast. Importantly, our downstream analysis models are robust to small inter-scan

intensity differences and reliably detect tissue boundaries independent of absolute signal levels.

To ensure that all fetal brains were included whole, the images were cropped to $160 \times 160 \times 160$ voxels around the centre of the brain. To align the brain to a pre-defined coordinate space, each scan was processed using the deep learning-based BEAN tool[56]. This convolutional neural network automatically localises the brain in the image and outputs seven affine transformation parameters (3 rotation, 3 translation, and 1 isotropic scale factor) to align the scan rigidly.

At each stage, QC was performed to ensure that the processing steps did not visibly distort the brain scans.

**Atlas Structural Parcellation Protocol.** As there is no established protocol for labelling US-derived structures in the fetal brain, we present the first US protocol for doing so using knowledge from previous fetal MRI parcellations maps[8,15,68,69] and histology studies[70].

We performed parcellation on nine weekly US atlases between 18 and 27 weeks' gestation, and included structures that were clearly visible in these atlases, as described previously[5]. The atlases were derived from a subset of the INTERGROWTH-21$^{st}$ FGLS cohort, which was found to be demographically representative of the full FGLS cohort in terms of maternal characteristics, fetal sex and study site (see Supplementary Tables 3–5 in [5]).

Segmentation was performed on ITK-snap[71] by M.K.W, a researcher with over 6 years' experience in fetal brain US imaging, and verified by A.I.L.N. For structures previously segmented in MRI fetal atlases (i.e., CoP, CSP, CB, Th and BS), we used the Computational Radiology Laboratory atlas as a guide for our segmentation maps[15]. For structures not previously described by MRI (i.e., LV, ChP and FH), MKW followed the tissue boundaries and referred to histology reports[70] to delineate the structure accurately. The DGM label encompassed the hippocampus, putamen, caudate, amygdala, internal capsules and ventricular zone. The labelling was performed iteratively, with each structure inspected and updated in the axial, coronal and sagittal planes. Thus, the cortical and subcortical structures manually labelled on each atlas were: *CoP, WM, DGM, CB, Th, LV, ChP, FH, BS, and CSP*.

CoP lobe atlas parcellation (i.e., TL, OL, IL,PL,FL) was performed by aligning the CoP atlas masks to the MRI-derived Computational Radiology Laboratory cortical lobe parcellations[15] (see Supplementary Table 9 for description of sulcal regions included within each lobe demarcation), and mislabelled regions were identified by visual inspection and manually corrected (by M.K.W).

The structures and lobe parcellations were labelled in both hemispheres on the atlas. Figure 2 shows an example at 22 weeks' gestation.

**Population Structural Labelling.** The atlas-based segmentation maps were propagated to the subset of the FGLS fetal brain scans from the atlas cohort (number of scans = 404) using the nonlinear registration fields learnt in atlas creation[5]. This inclusion was based on GA at scan and availability of registration fields. The propagated labels were then inspected and manually refined by M.K.W, when required. In practice, 46 of the 404 propagated scans required correction (16 substantial, 30 minor), yielding a curated labelled set that combined atlas propagation with expert manual review.

The refined labelled dataset was divided into two subsets: $\mathscr{D}_{training}$ for model training and $\mathscr{D}_{testing}$ for evaluation. The split was performed at the subject level, allocating 80% of the data to training (334 scans). The split was randomised while preserving the distribution of scans across gestational weeks and visible hemisphere. The resulting subsets were found to be evenly balanced with respect to study site and newborn sex (Supplementary Table 10).

To perform segmentation, two DL algorithms were trained in a supervised fashion using $\mathscr{D}_{training}$. First, in order to measure TBV, the whole brain was extracted using a U-Net[72]. This network achieved an average Dice overlap of $0.96 \pm 0.01$ on a previously unseen $\mathscr{D}_{testing}$.

This Dice value refers specifically to the whole-brain extraction step, and not the individual anatomical labels.

A key challenge in segmenting structures in both 2D and 3D US scans is that visibility of complete tissue boundaries is hampered by the presence of strong acoustic shadows. Consequently, conventional segmentation tools result in topological mistakes, such as isolated segments, gaps, or holes in regions of poor contrast or occlusions. This necessitates manual or semi-automated post-hoc corrective steps to produce complete surfaces for extracting brain measures (e.g., lobe volumes). Here, we extracted topologically correct segmentations of all 10 brain regions from each 3D US volume using the Topology-Enforcing Diffeomorphic Segmentation Network (TEDS-Net)[9,10].

In brief, TEDS-Net is a convolutional neural network that warps a pre-defined set of binary masks (i.e., the priors) to match each target structure of the imaged brain. Each initial binary mask is constructed with the desired topological properties of the structure of interest. In this work, we used the atlas labels at 22 weeks' gestation as the prior shape for each structure. The warping process involves performing a continuous diffeomorphic deformation that resamples the initial binary mask to its corresponding structure within each brain scan. This constrained deformation provides direct topological guarantees by explicitly discouraging the introduction of gaps on the surfaces of the target anatomies.

TEDS-Net was used to directly segment the 10 cortical and subcortical regions delineated on the atlases: CoP, WM, DGM, CB, Th, LV, ChP, FH, BS, and CSP. On the unseen $\mathscr{D}_{testing}$, it achieved an average Dice overlap of $0.78 \pm 0.03$ across these structures, with per-structure scores provided in Supplementary Fig. 9. Visual inspection of the test dataset's segmentation masks further confirmed that no manual refinement was necessary.

Lobar parcellations (FL, TL, PL, OL, IL) were generated by warping the 22 weeks' gestation atlas-based parcellation templates to each individual scan using the continuous diffeomorphic deformation learnt in TEDS-Net. As these lobar labels were derived from propagated templates rather than independently hand-delineated ground truth, per-lobe Dice coefficients are not reported. Dice metrics shown in Supplementary Fig. 9 therefore refer only to the manually delineated anatomical structures.

To ensure segmentation accuracy and exclude hallucination artifacts, we used a two-tier QC strategy: 1) manual review of a representative 2% test subset (≈70 scans) spanning gestational ages and image qualities, and 2) validation on an independent dataset of 90 fetal brain volumes with paired same-day MRI, where US-derived labels closely matched MRI-derived segmentations[51]. Together, these confirm anatomically reliable outputs.

Another challenge in US brain image analysis is that the increasingly calcifying skull bones and their convex shape create an acoustic cavity that, when interacting with the sound waves, cause shadows and reverberation artifacts. These limit the visualisation of both cerebral hemispheres in the brain volumes. As such, TEDS-Net yielded segmentation masks only for the visible hemisphere (distal to the US probe), shown in Table 1. To achieve this, distal hemisphere was used as an input into TEDS-Net, to ensure the correct side was labelled. Hemisphere labelling was performed using a deep learning classifier which achieved an accuracy of 99.9% on the held-out test set. In our dataset, more right hemispheres were imaged at sufficient quality than left (Supplementary Table 3 and Supplementary Table 10), consistent with previous reports that fetuses more often adopt a rightward head position at this gestational age range[73]. Consequently, each 3D US volume produced one value per brain measure.

**Image derived phenotypes (IDPs).** From the trained segmentation models, masks and volumes were computed for the entire FGLS population. For example, once the CoP voxels had been segmented from each 3D scan, the global size measures, i.e., volume (V) and

surface area (SA), were computed from each segmentation mask generated by TEDS-Net, $\mathbf{Y}_i \in \mathbb{R}^3$, from image $i$. CoPV was calculated by summing the voxels ($\mathbf{v}$) labelled as CoP (i.e., $\text{CoPV} = \alpha \sum_{\mathbf{v} \in S} \mathbf{v}$, where $S = \{\mathbf{v} : \mathbf{Y}_i = 1\}$ and $\alpha$ is the voxel resolution (i.e., isotropic, 1 voxel = 0.6 mm$^3$)). To facilitate the computation of SA, the CoP segmentation mask was converted into a triangular mesh $\mathscr{M}_i$ using the Lewiner marching cube algorithm ($f_L$), which, crucially, also preserves the topology of the underlying structure[74]. CoPA was estimated by summing the areas of all triangular faces of the mesh.

CoPD was defined as the closest distance between the skull boundary and each position within the CoP. To measure the distance, we computed the Euclidean distance transform of the TBV label and recorded the distances at each position within the CoP label. Mathematically this can be written as:

$$\mathbf{CoPD}_i = \text{dist\_transf}(\mathbf{TBV}_i) \times \mathbf{CoP}_i \tag{1}$$

$$\text{dist\_transf}(\mathbf{TBV}_i) = \min_{p \in \mathbf{TBV}_i} \sqrt{\left(q_x - p_x\right)^2 + \left(q_y - p_y\right)^2 + \left(q_z - p_z\right)^2} \tag{2}$$

where $q$ represents the locations of background voxels and $p$ are the locations within TBV.

CoPT was measured at each point along the cortical surface by computing the orthogonal projection from the surface to the nearest boundary point along the local gradient vector, inspired by previous voxel-wise techniques for estimating CoPT in MRI[75,76]. This voxel-wise distance transform approach achieves the same goal as MRI-based cortical thickness estimation but uses a different implementation since surface-based potential field models commonly applied in MRI cannot be used reliably in ultrasound. To validate these measures, we confirmed that CoPT estimates were within the 1.5-3.0 mm range reported in MRI atlases for 18-27 weeks' gestation[16,52] and showed consistent age-related trends across the cohort.

To generate growth standards for brain regions of functional relevance, the CoPV and average CoPT and CoPD was found at each parcellation lobe. To approximate SFD, instead of measuring the average depth in the insula, with incorporates both the inner and outer boundaries, we measured the median depth in the insula.

## MRI atlas comparison

To validate the structural labelling, the atlas labels were compared to five publicly available, labelled MRI atlases, spanning a similar gestational age range[8,15–19]. An overview of each atlas used is provided in Supplementary Table 11. As different structures are labelled across the atlases, ranging between 4 and 124 labels, a common set of labels was created by combing the available structures (described in Supplementary Table 12 and shown in Fig. 4a). In US images, the FH, LV and ChP have clear tissue boundaries; however, the distinction between structures is lost in MR images resulting in this ventricular system (VS) being segmented as one structure. Thus, to compare the VS volume across the five atlases, the US-derived FH, LW and ChP labels were combined across both hemispheres and the volume of each structure computed.

## Fetal brain maturation index

To compare biological and gestational ages from the IDPs, we largely followed the methodology presented by Smith et al[30]. for the prediction of Brain Age Delta from adult brain MRI data from the UK Biobank. All subsequent procedures used three-fold cross-validation, split at the subject level, with 1938 subjects used for training and validation and 968 held-out subjects used for testing.

We used the 28 extracted IDPs to construct a feature matrix $X$. The primary goal was to predict gestational age from this feature matrix,

denoted as $\hat{y} = f(\mathbf{X})$, where $f$ represents the predictive model. Each IDP was normalised to have zero mean and unit variance[77]. Subsequently, confounding variables were regressed out from the IDPs, following the approach detailed by Alfaro-Almagro et al. (2021)[78]. In our dataset, the set of identified confounds included sex, study site, and visible hemisphere. Notably, age-dependent confounds could not be removed due to their intrinsic relationship with the target variable.

Each confound was numerically encoded and then demeaned. These confounds collectively regressed out from the feature matrix $\mathbf{X}$ using a linear model without an intercept (as all confound variables were demeaned):

$$\hat{X} = X - VV^+ X \tag{3}$$

Here, $V$ is the matrix of demeaned confounds, and $V^+$ denotes the Moore-Penrose pseudo-inverse of $V$. This regression-based unconfounding was performed simultaneously for all confound variables together.

To quantify the influence of each individual confound, we calculated the percentage variance explained (%$VE$) and the percentage unique variance explained (%$UVE$). Specifically, for a given confound $V_1$, we define the predicted component of $X$ explained solely by this confound (if acting alone) as:

$$\hat{X}_1 = V_1 V_1^+ X \tag{4}$$

where $V = \begin{bmatrix} V_1 V_n \end{bmatrix}$ is the matrix of all confounds and $V_n$ are the remaining confounds. The percentage variance explained by $V_1$ for each IDP (column $j$ in $\mathbf{X}$) was then computed as:

$$\%VE_j = 100 \times \frac{\hat{X}_{1,j}^T \hat{X}_{1,j}}{X_j^T X_j} \tag{5}$$

Similarly, the percentage unique variance explained (%$UVE$) for each IDP (column $j$), representing the variance explained by confound $V_1$ not already accounted for by other confounds $V_n$, was computed as:

$$\%UVE_j = 100 \times \frac{X_j^T X_j - \hat{X}_{n,j}^T \hat{X}_{n,j}}{X_j^T X_j} \tag{6}$$

where $\hat{X}_{n,j} = V_n V_n^+ X_j$ represents the adjusted component of feature $j$ explained only by the remaining confounds $V_n$. Although each confound individually accounted for a small portion of the variance in $X$ (sex: 0.485%, hemisphere: 1.71%, site: 1.28%), all were removed to avoid arbitrary thresholding in confound selection.

The fully normalised and deconfounded feature matrix $\hat{X}$ served as input to a Random Forest model[79] to regress gestational age, implemented using the Scikit-learn implementation[80]. The model comprised 100 estimators with a maximum feature ratio of 0.15.

To identify the most informative IDPs contributing to the fetal brain maturational index, we examined feature importance measures derived from the Random Forest model. This included the mean and standard deviation of impurity decreases within each tree. Additionally, we used SHAP (Shapley Additive exPlanation) values[81] to assess the impact of each feature on individual predictions, thereby evaluating the consistency of feature importance across the gestational age range.

## Statistical analysis

To investigate the relationship between specific brain volumes (TBV and all 15 brain regions) and gestational age, regional brain growth was modelled as a function of gestational age. To account for the increasing

inter-subject variability with gestational age, the mean and standard deviation (SD) were modelled separately using fractional polynomial regression within the GAMLSS framework in R. This approach also allows for modelling of higher-order distributional parameters, such as skewness and kurtosis, thereby accommodating non-Gaussian distributions and providing robust estimations of centiles.

Based on our previous publications and WHO recommendations for comparing the similarities in growth across populations[20,21,62] we used variance component analysis to calculate the percentage of total variance due to between-study site differences for all brain measures. Variance components were calculated by nonparametric generalised linear mixed models (using the R lme4), which included as covariates sex and gestational age at assessment. These variables were included as fixed effects with study site as random effect. The method was chosen because it assumes a discrete instead of a Gaussian distribution of the random effects and showed better fit than Maximum Likelihood and Restricted Maximum Likelihood models according to the lower Akaike information and Bayesian information criteria.

In addition, for each brain measure, SSDs were calculated as the difference between the mean from a given site and the mean of all sites together, expressed as a proportion of the SD of the pooled data across sites for each measure. As pre-specified in the INTERGROWTH-21[st] Project protocol and recommended by WHO[21], SSD values in the range of ± 0.5 units of the pooled SD were taken as adequate to determine whether the data from all sites could be pooled.

### Infant follow-up

Across all sites, standardised clinical care and feeding practices were implemented using the INTERGROWTH-21[st] Neonatal Group protocols (www.intergrowth21.org.uk). Exclusive breastfeeding up to 6 months and appropriate nutritional support for infants born preterm were promoted during and after pregnancy, as recommended[82]. Detailed information was obtained from the mother at age 1 and 2 about the infant's health, severe morbidities, hospitalisations, length of breastfeeding, timing of the introduction of solid food, age at weaning, feeding practices and food intake, using specially produced forms (www.intergrowth21.org.uk). The proportion of infants receiving breast milk, and vitamin and mineral supplements, and those following a special diet were estimated at age 1 and 2. Similarly, at age 1 and 2, the infant's weight, length and head circumference were measured following WHO protocols, and their age- and sex-specific z-scores and centiles were compared to the WHO Child Growth Standards[21]. These anthropometric measures, as indicators of general nutrition at age 2, are strongly predictive of later attained height, development and human capital[42].

For the neurodevelopmental evaluation, children were scheduled for assessment at 2 years of age in five of the eight original sites: Nagpur, Nairobi, Oxford, Pelotas and Turin using tools specifically developed or selected for this purpose[83]. The sites in Beijing, Muscat and Seattle did not participate because of logistical and administrative reasons, delays in the start of the study and/or staff availability.

We assessed neurodevelopment using the INTERGROWTH-21[st] Neurodevelopmental Assessment (INTER-NDA; www.inter-nda.com), an international, psychometrically valid and reliable, standardised tool, targeted at children aged 22–30 months, which measures multiple dimensions of early development using a combination of directly administered, concurrently observed and caregiver reported items[83]. It was designed to be implemented by non-specialists across multinational settings, and includes a reduced number of culture-specific items comprising six domains measuring cognition, language, fine and gross motor skills, and positive and negative behaviour, in an assessment time of 15 min on average.

The INTER-NDA has been validated against the Bayley Scales of Infant Development III edition[84]. Based on established guidelines[85], it showed 'good' agreement with interclass coefficient correlations across domains ranging between 0.75 and 0.88. Its norms are the first international standards of early child development, constructed according to the prescriptive WHO approach using data from five of the eight original INTERGROWTH-21[st] study sites[12]. To date, over 40,000 children in 26 countries have been assessed using the INTER-NDA.

Vision was assessed using the Cardiff Visual Acuity and Contrast Sensitivity tests for binocular vision[13]. These are indicative of the integrity of the central visual pathway, and as directly observed neurodevelopmental markers, are unlikely to be affected by cultural influences and co-occurring disturbances in cognitive, hearing and language skills.

Motor development was assessed against four WHO milestones that are less likely to be affected by recall bias: sitting without support, hands-and-knees crawling, standing alone, and walking alone[86]. Trained staff collected the data on a form with pictures of the relevant child positions and corresponding definitions. Parents were asked to report the age in months and weeks when they "first observed" or "never observed" the milestones. We assessed the age (in months) at which WHO gross motor milestones were first achieved.

All INTER-NDA assessors were trained and standardised centrally. All assessors were subject to a protocol adherence and reliability assessment following training; only those with protocol adherence scores in excess of 90% and inter-rater reliability of >0.8 conducted assessments. The administration of the above tests was supported by a tablet-based data collection and management system. Field staff were unaware of the INTER-NDA domain and total scores for individual children and sites. Data were uploaded onto secure servers as soon as each assessment was completed and compared to the international normative values established by the INTERGROWTH-21[st] Project[12].

### Reporting summary

Further information on research design is available in the Nature Portfolio Reporting Summary linked to this article.

## Data availability

The labeled ultrasound atlases will be made publicly available on https://intergrowth21.com/research/brain-atlas-project. The derived growth curves are available in the Supplementary materials. Owing to the data still being under analysis for the principal and secondary objectives of the study protocol, anonymized image data will be made available on reasonable request for academic use only and within the limitations of the informed consent. Requests must be made to the corresponding author or to the INTERGROWTH-21st Consortium at intergrowth@wrh.ox.ac.uk. Full conditions of access are available in the INTERGROWTH-21st study protocol at (https://intergrowth21.com/research/brain-atlas-project). Every request will be reviewed by the INTERGROWTH-21st Consortium Executive Committee with due promptness. After approval, the researcher will need to sign a data access agreement with the INTERGROWTH-21st Consortium. Source data are provided with this paper.

## Code availability

The complete data analysis pipeline is available here: (https://oxford-omni-lab-org.github.io/OMNI_ultrasound/NormativeGrowthTrajec/index.html). All ultrasound processing was performed on Python 3.11, with the deep-learning implementation using Pytorch 2.1.0. Statistical analysis was performed in R-Studio (version 2024.04.2 + 764) using GAMLSS (version 5.4-22).

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

## Acknowledgements

M.K.W. and A.I.L.N. are supported by the Bill and Melinda Gates Foundation. M.K.W was also supported by the EPSRC Doctoral Prize Scheme [grant number EP/W524311/1]. M.F. is supported by a Clinical Research Training Fellowship (MR/V029169/2) funded by the Medical Research Council, UK. We would like to thank the Health Authorities in Pelotas, Brazil; Beijing, China; Nagpur, India; Turin, Italy; Nairobi, Kenya; Muscat, Oman; Oxford, UK and Seattle, USA, who facilitated the project by allowing participation of the study sites. We are grateful to Philips Medical Systems for the ultrasound equipment and technical assistance. We also thank MedSciNet U.K. Ltd for the website development and maintenance and support of the online data management system. We thank the participating hospitals, parents and infants who participated in the studies and the more than 200 members of the research teams who made this project possible. Full acknowledgment appears at (https://intergrowth21.com/research/intergrowth-21<sup>st</sup>). Luke Conaboy, Stephan Uphoff, Albert Alonso, Valentin Bacher, and the OMNI Lab provided invaluable help and advice.

## Author contributions

M.K.W., A.I.L.N, N.K.D, S.H.K and J.V. designed the study with input from M.F., A.T.P. and A.S. M.F., M.C., L.C.I., Y.J., M.G., W.Q., E.B., M.P., and F.C.B. oversaw data collection. The data were curated by M.K.W. and A.I.L.N. A.I.L.N., M.K.W., N.K.D., R.B.G. and E.O.O. analysed the data. A.I.L.N., M.K.W., N.K.D., and L.S.H. designed and implemented the image analysis algorithms, with input from M.J., S.S., and T.E.N. L.C.I., A.T.P., and A.W. coordinated the study. Z.M. assisted with the interpretation of the results. A.I.L.N., S.H.K. and M.K.W. wrote the initial drafts of the paper; JV expanded the initial draft and made it consistent with the INTERGROWTH-21<sup>st</sup> conceptual frame; all other authors including A.J.N. and Z.B., contributed and revised it critically for important intellectual content. All authors approved the final version for publication.

## Competing interests

J.A.N. and A.T.P. are Senior Scientific Advisers of Intelligent Ultrasound but the company has no financial or intellectual property links with the research described in this paper and the described work is entirely based on their academic work. We declare that all the other authors have no competing interests as defined by Nature Research, or other interests that might be perceived to influence the results and/or discussion reported in this paper. A.T.P. is supported by the Oxford Partnership Comprehensive Biomedical Research Centre with funding from the NIHR Biomedical Research Centre (BRC) funding scheme. The views expressed herein are those of the authors and not necessarily those of the NHS, the NIHR the Department of Health or any of the other funders. The remaining authors declare no competing interests.

## Additional information

¹Oxford Machine Learning in Neuroimaging Laboratory, Department of Computer Science, University of Oxford, Oxford, United Kingdom. ²Nuffield Department of Women's & Reproductive Health, University of Oxford, Oxford, United Kingdom. ³Department of Paediatrics, Institute of Developmental and Regenerative Medicine, University of Oxford, Oxford, United Kingdom. ⁴Oxford Maternal & Perinatal Health Institute, Green Templeton College, University of Oxford, Oxford, United Kingdom. ⁵School of Public Health, University of California, Berkeley, CA, United States of America. ⁶Department of Clinical Nutrition and Dietetics, College of Health Sciences, University of Sharjah, Sharjah, United Arab Emirates. ⁷Faculty of Epidemiology and Population Health, London School of Hygiene & Tropical Medicine, London, United Kingdom. ⁸Departments of Obstetrics and Gynecology and of Global Health, University of Washington, Seattle, WA, United States of America. ⁹Nagpur INTERGROWTH-21st Research Centre, Ketkar Hospital, Nagpur, India. ¹⁰School of Public Health, Peking University, Beijing, China. ¹¹Dipartimento di Scienze Pediatriche e dell'Adolescenza, SCDU Neonatologia, Universita di Torino, Turin, Italy. ¹²Department of Family and Community Health, Ministry of Health, Muscat, Sultanate of Oman. ¹³Department of Obstetrics and Gynaecology, Faculty of Health Sciences, Aga Khan University Hospital, Nairobi, Kenya. ¹⁴Post-graduate Course in Health in the Life Cycle, Universidade Católica de Pelotas, Pelotas, Brazil. ¹⁵Blavatnik School of Government, University of Oxford, Oxford, United Kingdom. ¹⁶African Health Research Institute, KwaZulu-Natal, South Africa. ¹⁷MRC/Wits Rural Public Health and Health Transitions Research Unit (Agincourt), School of Public Health, Faculty of Health Sciences, University of the Witwatersrand, Johannesburg, South Africa. ¹⁸Department of Engineering Science, University of Oxford, Oxford, United Kingdom. ¹⁹Department of Physiology, Anatomy and Genetics, University of Oxford, Oxford, United Kingdom. ²⁰Oxford Centre for Integrative Neuroimaging (OxCIN), FMRIB Centre, Nuffield Department of Clinical Neurosciences (NDCN), University of Oxford, JR Hospital, Oxford, United Kingdom. ²¹Australian Institute for Machine Learning (AIML), School of Mathematical and Computer Sciences, The University of Adelaide, Adelaide, Australia. ²²South Australian Health and Medical Research Institute (SAHMRI),

Adelaide, Australia. [23]Big Data Institute, Li Ka Shing Centre for Health Information and Discovery, Nuffield Department of Population Health, University of Oxford, Oxford, United Kingdom. [24]Center for Global Child Health, Hospital for Sick Children, Toronto, Ontario, Canada. [25]Institute for Global Health and Development at Aga Khan University, South-Central Asia & East Africa, Karachi, Pakistan. [26]These authors contributed equally: Madeleine K. Wyburd, Stephen H. Kennedy. [27]These authors jointly supervised this work: Jose Villar, Ana I. L. Namburete. ✉e-mail: madeleine.wyburd@cs.ox.ac.uk; ana.namburete@cs.ox.ac.uk

