## [Transparent Peer Review file · Nature Communications]

Normative growth trajectories of fetal brain regions validated by satisfactory maturation of neurodevelopmental domains at 2 years of age

Corresponding Author: Dr Madeleine Wyburd

Version 0:

Reviewer comments:

Reviewer #1

(Remarks to the Author)

General Impressions:

Overall, this is a really interesting paper that represents the results of a significant body of work. The real strength of this study is in the size of the dataset and the wide geographic/racial/ethnic sample size of the normative cohort.

The paper represents the result of a significant amount of image analysis work applied to a highly challenging dataset, and builds on previously published ultrasound atlases by the group.

One of the key findings is the consistency of basic regional brain growth trajectories in postnatally confirmed normal development

across very different populations around the world.

The results are clearly important, but also because of this, it is important that these findings are well supported by the study measurement procedures and the analyses of them.

Limitations of the work in terms of a wider scientific understanding of brain development are the relatively limited gestational age range (18-27 gestational weeks) and concern over apparent limitations in the accuracy and precision of a number of the measures included that potentially limit/detract from the general findings. These concerns could potentially be addressed by the inclusion of more information and the use of additional analysis procedures.

Methods:

Image resolution:

page 15: image resampling: it is mentioned that images needed to be resampled from a median pixel size using an interpolation scheme.

What was the actual range (max, min, standard deviation) of the pixel sizes in each dimension?

Did the resolution vary with other factors in the study: eg between site or gestational age of the fetus?

It appears the images were in fact of anisotropic sampling resolution, with some having a much finer pixel size than that used for analysis.

How did the anisotropic resolution vary wrt anatomical axis in the different datasets? (was it random or for example were all anatomically axial planes higher resolution?)

It is a concern that correctly reducing resolution (increasing pixel size) does not simply involve linear interpolation as this can induce aliasing artifacts because of pixel skipping.

Image Intensity normalisation:

It is mentioned on page 15 that the images were intensity normalized to have a range between 0 and 1. Does this mean the maximum intensity in the volume is now always 1? It is not clear what this means in terms of the resulting variation of the normalized signal in the tissues/ tissue boundaries. Conventionally in image analysis intensity normalization is achieved relative to some reference region of anatomy or tissue type. What is the resulting normalized signal level in (for example) the boundary between cortex and CSF? (which is important for labelling)

Methods

=====

Label Segmentation and quality control of anatomical labels:

A general concern is the use of deep learning methods for anatomical labelling that is then used as a quantitative measurement. It is important to note that these deep learning methods used (unlike conventional model based labelling methods) do not provide an indication of error or certainty in the labelling result that is created. Fundamentally (and particularly given the range of data quality and high variance in the some of the final results) it is not clear how the investigators ensure that the tissues are actually being measured or whether the labelling is simply being hallucinated by the network for images where identification of tissues is not possible. (given the network was trained on fetal brain labels, it can only produce results that look like patterns of fetal brain labels). There could in fact be a significant proportion of the images where no actual measurement was made in some or all brain regions. In these cases the network would potentially simply provide a guess rather than a measurement. There is apparently no procedure employed to address labeling accuracy or to detect hallucination of measurements. (no actual labelling was excluded, only initial image quality checks) This limitation needs to be explained in the paper and appropriate quality control procedures need to be employed for the automated labeling.

Segmentation Testing/Performance Measures:

It is a concern that the entire training and testing was based originally on only 9 actual manually delineated reference scans. On page 15 it states "We performed parcellation on nine weekly US atlases between 18 and 27 weeks gestation, and included structures that were clearly visible in these atlases, as described previously" These were then used (via an automated procedure) to create 334 training datasets with apparently some manual checking which needs to be described in more detail.

There is a lack of information about the DICE performance of the labeling on the actual regions for which results are presented. These DICE measures are apparently derived from 70 cases, but no information about the demographics of these scans is included. (age, gender, left or right hemisphere etc) or how it relates to the whole study demographics.

On page 16 it is stated that: "This network achieved an average Dice overlap of 0.96 ± 0.01 on a previously unseen test dataset" It is not clear which anatomical label(s) this summary DICE statistic refers to. On page 17 it is stated that: "TEDS-Net achieved an average Dice overlap across all structures of 0.78 ± 0.03 on the unseen test dataset, A full description of the network performance is shown in Suppl Fig. 7" Again this appears to be a summary average for some labels: it is not clear how they relate to the results measures.

Supplementary figure 7 shows DICE values for some whole brain regions, but should be changed to show the DICE values for each of the volumetric regions presented in the results (at least all the volume measures by lobe mentioned listed in Table 1). This figure should also indicate clearly the full range of the DICE values (it is not clear if the bars indicate standard deviation or range or what the other dots indicate). The low average values of DICE (Supplementary Fig 7) for relatively large regions such as the whole cortical plate (0.6 at 18GW) is a real concern. Information about the sub regions actually used (eg lobe) estimates that are presented in the main results (and their max and min) is clearly needed here as they may well be even worse.

Atlas Data Balance:

An important missing factor for the atlas/training subjects is what sub-groups/cohorts they were taken from:

do the atlas labels used to train the segmentation allow an representative/unbiased segmentation of the potentially different cohorts?

(full demographic information relating to the training and testing data should be provided: age/sex/race geographic area etc is needed and

how it relates to the same demographics across full study data)

This is particularly important if the authors intend the system to be used in other researchers that may have different cohort balances.

Given only one hemisphere is being measured the atlas data should also include hemispheric balance in relation to main demographic factors such as age/sex/race etc.

Results

=====

Atlas image comparison to MRI atlases: This is really useful!

It is stated that an age range of 18-27 (abstract) weeks was analyzed in the main cohort, but the comparison of the ultrasound atlas used for with MRI results (Figure 4) apparently only includes 18-26 gestational weeks.

This needs to be explained and/or corrected.

Presentation:

Page 16 states "As such, TEDS-Net yielded segmentation masks only for the visible hemisphere (distal to the US probe)." It is a concern that all the data presented is apparently only for one hemisphere and pooled (even given the arguments presented to support this).

The figures of labels and results (eg figure 2) give the impression the whole brain was analyzed.

The figures should be corrected to show the actual labelling scheme/region used for measurements.

It also needs to be explained how the hemisphere was chosen (particularly for images where there is some oblique angulation of the brain with the sensor).

It is argued that the hemispheric differences were small enough to ignore and data could be directly pooled.

Given the sometimes low performance of the segmentation on only 70 cases, and the high variance and outliers seen in many of the results graphs,

it would seem that stronger evidence is needed to support this decision for each of the measures over the entire developmental range studied.

Given the amount of data available it would seem reasonable to include (perhaps in the supplementary sections) separate left and right hemisphere growth data for each of the label measures used in the study.

To help support the accuracy of the measures of each region, the paper should provide the growth model trajectory fits (with confidence intervals) for both hemispheres of each label (not just percentiles) to confirm the left right consistency over the age range.

This would help support the claimed accuracy of all the results and allow comparison to earlier fetal brain studies.

All figure captions of results should clearly state estimates are based only one hemisphere measurement where appropriate and whether the

measure is doubled to match a full brain (volume) estimate or is just one hemisphere.

Fig 7 has no numbers indicating gestational weeks marked on the horizontal axis.

Variability in final measures:

Given the deep learning methods used for automated labeling do not provide an indication of certainty in the labelling results and

there was apparently an image QC procedure used for the entire dataset, the results do

exhibit a high variance and apparently have some very extreme outliers (eg supplementary figure 4 and 5)

An example is the basic measures of white matter volume normalized by total brain volume (which as large regions, so should be reasonably reliable) which appear to contain a number of cases where the ratio WM/TBV differs by a factor of 3 or more from the mean.

It is therefore unclear how reliable the labeling and quality control are for the label results (particularly the smaller more challenging regions).

Although the atlas volumes are compared to the MRI atlas volumes (eg figure 4), there is less discussion of how the actual results of the full study data relate to previously published MRI atlases for this age range.

This is particularly important for measures that appear to vary significantly from previous MRI studies (for example cortical plate thickness)

The cortical thickness results are a particular concern as they indicate a very large variability in the average estimates and appear to indicate there is no significant increase in cortical thickness over the age range (and even a reduction at early time points)

This is not mentioned or explained, and the results are perhaps not well supported by previous studies.

The thickness of the CP appears large for this age range (between 2 and 2.5mm or greater even at 19 or 20 weeks gestation).

There is no clear description of how CP thickness was calculated or validated so it is difficult to see how these results relate to previous findings.

Overall, given the large number of measures evaluated, it would perhaps be useful to focus the paper on those measures that are most accurately/precisely estimated from ultrasound and to not include those (for example cortical thickness) that are perhaps more questionable when estimated from ultrasound studies.

(Remarks on code availability)

I briefly looked at the github pages.

They contain instructions and test data to support use of the code.

The installation of the code was relatively complex, requiring additional libraries so I could not run and verify the code myself on my home laptop.

Given time and expertise in python library versions it seems feasible to get this running using the documentation provided.

Reviewer #2

(Remarks to the Author)

While the study offers several strengths—including a large, multi-centre cohort, state-of-the-art deep-learning segmentation, and the first normative 3D-ultrasound growth curves for 16 fetal brain structures—two limitations temper enthusiasm.

1) The manuscript presents only "prescriptive" trajectories derived from low-risk pregnancies, omitting a comparison arm of fetuses with known neurodevelopmental abnormalities. Given that abnormal brain development may follow divergent volumetric paths, the absence of such a contrast group precludes assessment of whether the reported percentiles retain discriminative validity at the pathological end of the spectrum. Consequently, the clinical incremental value over their 2023 study is limited.

2) The authors excluded 101 infants with 2-year INTER-NDA scores below the 3rd percentile to preserve the atlas's "normative" nature. However, the intracranial growth trajectories of these excluded infants should have been explicitly plotted against the retained cohort to quantify their deviation from normal centiles. If trajectories of the excluded cohort do not diverge significantly, the current imaging-derived phenotypes (e.g., rILV:rPLV) may lack sensitivity for early pathology; conversely, marked deviations would substantiate their utility as early biomarkers. This omission represents a missed opportunity for clinical validation.

Additional points:

1) The discussion claims that "observed differences ... are primarily due to socioeconomic, educational and class disparities, rather than genetic variants." This sweeping statement requires formal mediation analysis before it can be asserted with confidence.

2) The statistical section would benefit from review by an independent biostatistician, particularly to confirm the adequacy of the fractional-polynomial assumptions and the handling of repeated-measures correlation in the growth models.

(Remarks on code availability)

Reviewer #3

(Remarks to the Author)

This is a followup analysis of a previously published large-scale, multinational study based on fetal brain 3D ultrasound scans and advanced reconstruction methods. The authors build normative growth trajectories, introduce a novel "fetal brain maturation index" and propose the rILV:rPLV ratio as a potential functional correlate. The work is validated against neurodevelopmental outcomes at 2 years of age. There are some methodological and theoretical considerations that merit further attention and clarification:

1. The authors evaluated site effects using two strategies:

(1) a leave-one-site-out approach: it is unclear how the authors quantitatively assessed the statistical significance of

excluding each site. Could they provide the growth curves for each site individually to better visualize local deviations?
(2) the SSD (standardized site difference) metric: the metric merges within gestational windows, thereby ignoring potential site-by-age interactions (i.e., site effects on the developmental trajectory rather than a specific time point). It is also discussed by the authors that inconsistencies in mean gestational age disturbed the comparisons (line 169). Do the authors consider applying more modern nonlinear harmonization frameworks such as ComBat-GAM to more robustly address site and scanner heterogeneity.

In addition, the authors included site effect in the subsequent mixed-effects model. Were site effects statistically significant in these models?

2. While the study used a prescriptive, “healthy” cohort to build normative models, some individuals in the broader cohort had low neurodevelopmental scores were excluded from model construction. It would greatly strengthen the manuscript to test the brain maturation index on these children with suboptimal neurodevelopmental outcomes, or on individuals with diagnosed neurodevelopmental disorders, to evaluate the model’s generalizability and potential for early risk screening.

3. The authors presented the 3rd and 97th centiles as normative boundaries. However, certain metrics, for example FLT, PLT, and OLT in Fig. 7, show distributions that appear non-Gaussian at younger gestational ages. The 97th centile curve in particular seems to deviate far from the observed scatter, suggesting a potential misfit of the normal-based percentile approach. I suggest the authors to more formally examine distributional normality across gestational ages, and to consider alternative methods that would not assume Gaussianity, in order to derive more accurate and clinically meaningful centile boundaries.

4. Although the authors collected data up to 30 weeks’ gestation, the modeling focused only on 18–27 weeks. The rationale for selecting this window is well argued, but the exclusion of later scans might limit applicability for clinical follow-up into the third trimester. If data quality permits, the authors could consider extending their models toward the late second and third trimester in future work to provide more comprehensive reference trajectories.

5. The segmentation procedure was performed only on the distal hemisphere due to acoustic shadowing constraints. While a “distal hemisphere” confounder was included in the regression models, the study does not explicitly analyze potential lateralization or asymmetry effects, which could be biologically meaningful in early fetal cortical development.

6. The proposed rILV:rPLV ratio shows promise as an indicator of asynchronous cortical maturation, but the manuscript stops short of demonstrating its relationship with functional outcomes. Could the authors consider an exploratory analysis correlating rILV:rPLV deviations with neurodevelopmental scales to support its potential clinical utility?

(Remarks on code availability)

Version 1:

Reviewer comments:

Reviewer #1

(Remarks to the Author)

Overall, the authors have done well to clarify my misunderstanding and confusion and resolve the concerns I had. I have a couple of minor follow ups to the rebuttal that should be easy to address:

Contrast normalization:

With regard to the response to my question regarding intensity normalization (page 4), you could perhaps just make it simply clear in the response that. (I think that:?) unlike many standard MR image processing pipelines which often make use of a tissue intensity standardization (eg of CSF or white matter intensity) and/or a bias correction step, you make use of training across a large dataset with varying contrast, to deep-learn a contrast independent labeling and therefore negate the impact of potentially varying tissue/boundary contrast on the ultrasound segmentation process.

This seems to be the case? (if you are just making the maximum and minimum in the entire volume to be 1 and 0)?

Image Resolution:

On page 3 of the rebuttal in response to my question: “How did the anisotropic resolution vary wrt anatomical axis in the different datasets?”

I did not quite understand the answer (perhaps I just need something simpler?):

Beyond stating that a standard protocol was used to collect the slices: It was not clear from the response and the paper whether that meant the collected slices (with a given thickness and in plane resolution) were always orientated precisely

axially or did the anatomical orientation of the slices depend significantly on the fetal head orientation for each baby? i.e. Did the operator always manage to acquire approximately axial slices? Or was it very (or somewhat) baby + operator dependent? (do you have an approximate idea how big were the rotations away from precisely axial ?)

This is important because in your response you also mention that some datasets had a slice thickness down to 0.27mm while others had a slice thickness of up to 1.91mm. This is a considerable range (for example this is generally kept much more constrained in many MRI studies). As a result the slice orientation may impact the performance of the segmentation.

Cortical thickness:

Your new description of the calculation of cortical thickness should perhaps include a reference to the method? I assume it has been used previously? It also appears that the thickness was across the discretely labelled CP voxels (did not account for any sub-voxel partial volume effects) and therefore will only be accurate to +/- a voxel (?) at any given point? (even if the automated labeling is perfect). If so, you should make this limitation clear in the description.

(Remarks on code availability)

Reviewer #2

(Remarks to the Author)

Thank you for your reply, No more question.

(Remarks on code availability)

Reviewer #3

(Remarks to the Author)

This is an insightful study offering large-scale, multi-centered fetal brain data with comprehensive growth trajectories, providing a view of normative fetal brain development.

The authors have addressed my comments appropriately by providing supplementary statistics and by extensively discussing the possible limitations and future directions - the manuscript has been greatly improved.

(Remarks on code availability)

"Normative growth trajectories of fetal brain regions validated by satisfactory maturation of neurodevelopmental domains at 2 years of age"

REVIEWER COMMENTS

Reviewer #1 (Remarks to the Author):

General Impressions:

Overall, this is a really interesting paper that represents the results of a significant body of work. The real strength of this study is in the size of the dataset and the wide geographic/racial/ethnic sample size of the normative cohort. The paper represents the result of a significant amount of image analysis work applied to a highly challenging dataset, and builds on previously published ultrasound atlases by the group.

One of the key findings is the consistency of basic regional brain growth trajectories in postnatally confirmed normal development across very different populations around the world. The results are clearly important, but also because of this, it is important that these findings are well supported by the study measurement procedures and the analyses of them.

We thank the Reviewer for their positive and encouraging comments.

Limitations of the work in terms of a wider scientific understanding of brain development are the relatively limited gestational age range (18-27 gestational weeks) and concern over apparent limitations in the accuracy and precision of a number of the measures included that potentially limit/detract from the general findings. These concerns could potentially be addressed by the inclusion of more information and the use of additional analysis procedures.

The Reviewer has raised important concerns, which we shall address individually below.

Methods:

Image resolution:

page 15: image resampling: it is mentioned that images needed to be resampled from a median pixel size using an interpolation scheme. What was the actual range (max, min, standard deviation) of the pixel sizes in each dimension?

The 3D ultrasound volumes were resampled to an isotropic voxel size of 0.6 mm in each dimension, followed by centre-cropping to 160×160×160 voxels. For reference, the original scans had median dimensions of 444×254×111 voxels (IQR: 59×88×36), with a corresponding median voxel size of [0.32, 0.51, 0.85] mm (IQR: [0.05, 0.15, 0.28] mm) across the three axes. Specifically, the minimum, maximum, and standard deviation were 0.14, 0.76, and 0.04 mm in the x-dimension; 0.30, 1.01, and 0.15 mm in the y-dimension; and 0.27, 1.91, and 0.23 mm in the z-dimension. This distribution captures the range of variability in acquisition resolution across sites. These preprocessing steps follow the same protocol we have previously used in related studies^{1,2}. We have now included this information in the Methods:

“Across the dataset, voxel sizes ranged from 0.14-0.76 mm (SD 0.04) in the x-dimension, 0.30-1.01 mm (SD 0.15) in the y-dimension, and 0.27-1.91 mm (SD 0.23) in the z-dimension. This interpolation approach avoids pixel skipping and minimises the risk of aliasing artifacts. Given that most acquisitions had finer in-plane resolution and only modestly coarser through-plane resolution, the resampling primarily involved upsampling rather than downsampling, thereby further reducing the likelihood of aliasing” (lines 475-8).

Did the resolution vary with other factors in the study: eg between site or gestational age of the fetus?

Voxel resolution was determined by the ultrasound system settings and acquisition protocol, which were standardised across all sites and gestational ages as part of the INTERGROWTH-21st protocol. Accordingly, we did not expect systematic variation by site or gestational age. Consistent with this, the observed voxel size distributions showed only minor variability (median [0.32, 0.51, 0.85] mm; IQR [0.05, 0.15, 0.28] mm), reflecting equipment-related rather than biological or site-specific differences. We further examined voxel resolution against imaging site and gestational age and found no systematic differences. For clarity, we have added these plots to the Supplementary Information and noted this cross-site harmonisation in the Methods:

“Voxel size distributions were highly uniform across imaging sites and gestational ages, indicating effective cross-site harmonisation of acquisition protocols (Suppl. Fig. 8) (lines 481-3).

It appears the images were in fact of anisotropic sampling resolution, with some having a much finer pixel size than that used for analysis. How did the anisotropic resolution vary wrt anatomical axis in the different datasets? (was it random or for example were all anatomically axial planes higher resolution?)

The original 3D ultrasound acquisitions were indeed anisotropic, with voxel sizes typically finer in the in-plane dimensions (x–y) than the through-plane (z) dimension. This reflects the intrinsic properties of ultrasound imaging, where lateral and elevational resolutions are determined by the transducer beam profile, and the slice thickness is governed by mechanical sweep increments. Importantly, this anisotropy was systematic and consistent across sites, as acquisition protocols were standardised in the INTERGROWTH-21st study, and it did not vary with fetal orientation or gestational age (Suppl Fig. 8).

We also note that anisotropy in ultrasound acquisitions is generally less pronounced than in fetal MRI, where slice thickness can be several times larger than the in-plane resolution. In contrast, our dataset had voxel dimensions on the order of [0.32, 0.51, 0.85] mm (median, with relatively narrow IQRs), yielding near-isotropic resolution in most cases. To mitigate residual differences, all images were resampled to an isotropic voxel size of 0.6 mm³ prior to analysis.

It is a concern that correctly reducing resolution (increasing pixel size) does not simply involve linear interpolation as this can induce aliasing artifacts because of pixel skipping.

The Reviewer has raised a valid concern. To clarify, all images were resampled to an isotropic voxel size of 0.6 mm³ using trilinear interpolation, rather than pixel skipping, which avoids aliasing-related artifacts. In practice, the vast majority of our acquisitions had smaller voxel sizes than 0.6 mm in at least one dimension (median voxel size [0.32, 0.51, 0.85] mm), so the resampling involved modest *upsampling* in the through-plane direction and only minimal downsampling in the in-plane directions. As such, the risk of aliasing was very low. Furthermore, this resampling strategy is widely used in both ultrasound and MRI neuroimaging preprocessing pipelines, including our previous work. We have now stated this in the Methods:

“Across the dataset, voxel sizes ranged from 0.14-0.76 mm (SD 0.04) in the x-dimension, 0.30-1.01 mm (SD 0.15) in the y-dimension, and 0.27-1.91 mm (SD 0.23) in the z-dimension. This interpolation approach avoids pixel skipping and minimises the risk of aliasing artifacts. Given that most acquisitions had finer in-plane resolution and only modestly coarser through-plane resolution, the resampling primarily involved upsampling rather than downsampling, thereby further reducing the likelihood of aliasing” (lines 475-81).

Image Intensity normalisation:

It is mentioned on page 15 that the images were intensity normalized to have a range between 0 and 1. Does this mean the maximum intensity in the volume is now always 1?

The raw ultrasound data were stored as 8-bit images, with intensities in the range 0–255. For preprocessing, we performed a per-volume min–max normalisation, linearly rescaling the intensity values so that the minimum voxel value in each volume was set to 0 and the maximum to 1. Thus, after normalisation, the maximum intensity within each individual volume is indeed 1, and the minimum 0. This is a standard step in ultrasound and neuroimaging preprocessing pipelines, designed to account for differences in scanner gain and acquisition conditions, and does not alter the relative contrast structure within the volume. This has now been included in the Methods:

“...using per-volume min-max scaling, such that the minimum voxel intensity was set to 0 and the maximum to 1, thereby standardising all inputs while preserving relative contrast” (lines 484-6).

For completeness, we also computed the distribution of raw maximum intensities, which had a mean of 237 (SD 23, IQR 26) across the dataset and shown in *Suppl. Fig. 8*.

It is not clear what this means in terms of the resulting variation of the normalized signal in the tissues/ tissue boundaries. Conventionally in image analysis intensity normalization is achieved relative to some reference region of anatomy or tissue type. What is the resulting normalized signal level in (for example) the boundary between cortex and CSF? (which is important for labelling)

The per-volume min-max normalisation approach does not alter the relative contrast between tissue classes within the same volume; it simply maps the dynamic range of the scan to a common scale. For example, the contrast between cortical plate and CSF remains unchanged. The absolute normalised intensity of a given structure (e.g. cortex vs. CSF) may vary slightly across scans depending on acquisition gain and depth-dependent attenuation, which are inherent to ultrasound. However, our segmentation framework does not rely on absolute intensity thresholds; instead, it leverages deep learning models trained across thousands of scans to learn robust tissue boundaries despite inter-scan variability.

In summary, the normalisation ensures consistency across scans while preserving within-scan contrast, which is the critical property for reliable labelling. We have added this explanation to the Methods to make this point explicit:

“Per-volume min-max normalisation results in minor variation in the absolute normalised intensity values of specific tissues across scans, but preserves the relative intra-volume contrast (e.g., between cortical plate and CSF). Importantly, our downstream analysis models are robust to such inter-scan intensity differences and reliably detect tissue boundaries independent of absolute signal levels” (lines 486-91).

Methods

Label Segmentation and quality control of anatomical labels:

A general concern is the use of deep learning methods for anatomical labelling that is then used as a quantitative measurement. It is important to note that these deep learning methods used (unlike conventional model based labelling methods) do not provide an indication of error or certainty in the labelling result that is created.

Fundamentally (and particularly given the range of data quality and high variance in the some of the final results) it is not clear how the investigators ensure that the tissues are actually being measured or whether the labelling is simply being hallucinated by the network for images where identification of tissues is not possible. (given the network was trained on fetal brain labels, it can only produce results that look like patterns of fetal brain labels). There could in fact be a significant proportion of the images where no actual measurement was made in some or all brain regions. In these cases the network would potentially simply provide a guess rather than a measurement. There is apparently no procedure employed to address labeling accuracy or to detect hallucination of measurements. (no actual labelling was excluded, only initial image quality checks) This limitation needs to be explained in the paper and appropriate quality control procedures need to be employed for the automated labeling.

We agree that deep learning-based segmentation methods, unlike model-based approaches, do not inherently provide an uncertainty estimate, and that quality control (QC) is therefore critical.

As noted in the Discussion, our novel segmentation methods were thoroughly tested and verified on a representative subset (2% of the 4,205 scans). This subset was carefully chosen to reflect the full range of gestational ages, demographics and image qualities in the dataset. Each segmentation was compared to an expert-labelled ground truth (generated by atlas-based label-propagation and then manually inspected), providing strong evidence of the network's ability to generalise. While it was not feasible to perform manual QC across the entire dataset given its size (n= 4,205), this approach represents a pragmatic balance between methodological rigour and feasibility.

To clarify further, we have expanded the Methods to explain how atlas labels were generated and corrected, and how segmentation accuracy was evaluated on the test set. These labels were manually inspected and iteratively refined by experienced fetal neuroimaging researchers, ensuring their reliability as a benchmark.

“The atlas-based segmentation maps were propagated to the subset of the FGLS fetal brain scans (number of scans = 404) using the nonlinear registration fields learnt in atlas creation¹. This inclusion was based on GA at scan and availability of registration fields. The propagated labels were then inspected and manually refined by MKW, when required. In practice, 46 of the 404 propagated scans required correction (16 substantial, 30 minor), yielding a curated labelled training set that combined atlas propagation with expert manual review” (lines 529-35).

In addition, the same segmentation framework has been applied to an independent dataset of 90 fetal brain volumes that included same-day paired MRI acquisitions³. In this dataset, segmentation outputs from 3D ultrasound were validated against MRI-derived labels of the corresponding structures, providing concrete evidence that the model does not “hallucinate” anatomy but accurately delineates tissues even when modalities are compared.

We have revised the manuscript to make these points explicit in both the Methods and the Discussion:

“To ensure segmentation accuracy and exclude hallucination artifacts, we used a two-tier QC strategy: 1) manual review of a representative 2% test subset (≈ 70 scans) spanning gestational ages and image qualities, and 2) validation on an independent dataset of 90 fetal brain volumes with paired same-day MRI, where US-derived labels closely matched MRI-derived segmentations⁵⁶. Together, these confirm anatomically reliable outputs” (lines 576-81).

“Although DL-based segmentation could in principle hallucinate anatomy in poor-quality scans, we did not observe this. Manual review of 70 representative scans and independent validation in 90 volumes with same-day MRI demonstrated anatomically faithful labels, supports the reliability of our approach⁵⁶” (lines 400-3).

By including these sentences, we feel we have acknowledged that, while hallucination artefacts cannot be fully excluded, the QC performed and the consistency across two independent datasets (one including MRI ground-truth) strongly mitigate this concern.

Segmentation Testing/Performance Measures:

It is a concern that the entire training and testing was based originally on only 9 actual manually delineated reference scans.

On page 15 it states "We performed parcellation on nine weekly US atlases between 18 and 27 weeks gestation, and included structures that were clearly visible in these atlases, as described previously". These were then used (via an automated procedure) to create 334 training datasets with apparently some manual checking which needs to be described in more detail.

We thank the Reviewer for highlighting these points. While it is correct that atlas construction was initiated from nine manually delineated reference scans (covering 18–27 weeks' gestation), these were not the sole source of training data. Instead, they served as seeds for atlas-based label propagation: a well-established approach in neuroimaging (e.g., FreeSurfer and related MRI-based atlas propagation methods).

From these initial references, labels were propagated to 404 additional scans. Crucially, all propagated labels were manually inspected, and corrections were made where needed. Specifically, 46 scans required manual refinement (16 with substantial adjustments, 30 with minor corrections), resulting in a carefully curated labelled training set. This process was described in the Methods, but we have now revised the text to clarify the proportion of scans that were manually refined and clarified how the training/testing dataset was selected:

“The atlas-based segmentation maps were propagated to the subset of the FGLS fetal brain scans (number of scans = 404) using the nonlinear registration fields learnt in atlas creation 1. This inclusion was based on GA at scan and availability of registration fields. The propagated labels were then inspected and manually refined by MKW, when required. In practice, 46 of the 404 propagated scans required correction (16 substantial, 30 minor), yielding a curated labelled training set that combined atlas propagation with expert manual review” (lines 529-35).

“The refined labelled dataset was divided into two subsets: $D_{training}$ for model training and $D_{testing}$ for evaluation. The split was performed at the subject level, allocating 80% of the data to training (334 scans). The split was randomised while preserving the distribution of scans across gestational weeks and visible hemisphere. The resulting subsets were found to be evenly balanced with respect to study site and newborn sex (Suppl. Table 10).” (lines 536-41).

Finally, we already indicate that our atlas-derived labels were compared to existing MRI-based fetal brain atlases and showed strong correspondence, further supporting their anatomical validity. Together, this demonstrates that the labelled training dataset was not solely reliant on the nine initial manual segmentations, but on a larger and rigorously quality-controlled set.

There is a lack of information about the DICE performance of the labeling on the actual regions for which results are presented. These DICE measures are apparently derived from 70 cases, but no information about the demographics of these scans is included. (age, gender, left or right hemisphere etc) or how it relates to the whole study demographics.

We thank the Reviewer for this useful suggestion. The Dice performance metrics were indeed derived from 70 representative cases spanning the study gestational age range and variability in image quality.

To improve transparency, we have now compiled demographic and cohort information for both the training and test datasets, including age, sex, study site, and visible hemisphere. This allows assessment of how representative the atlas-labelled subset is relative to the full study cohort. We have included the table below in the revised manuscript (Supplementary Table 10), enabling readers to evaluate cohort balance and potential biases.

Supplementary Table 10

Visible Hemisphere	Train		Test	
	Left	Right	Left	Right
Newborn Sex				
Male	70	111	17	22
Female	49	104	11	20
Study site				
Beijing, China	19	43	5	8
Muscat, Oman	25	42	5	9
Nagpur, India	18	23	3	3
Nairobi, Kenya	14	31	3	10
Oxford, UK	16	29	4	3
Pelotas, Brazil	8	14	1	3
Seattle, USA	7	11	1	2
Turin, Italy	12	22	6	4
Gestational week				
18	25	25	5	6
19	16	17	1	6
20	15	30	3	7
21	14	24	4	4
22	8	15	1	4
23	12	37	4	6
24	10	19	5	1
25	6	20	0	5
26	13	28	5	3

On page 16 it is stated that: "This network achieved an average Dice overlap of 0.96 ± 0.01 on a previously unseen test dataset". It is not clear which anatomical label(s) this summary DICE statistic refers to.

The reported Dice coefficient refers specifically to whole-brain extraction (used to measure TBV), performed with a U-Net model. As described in the Methods, this network achieved an average Dice overlap of 0.96 ± 0.01 on a previously unseen test dataset ($n = 70$ scans). Nevertheless, we have revised the text to make clear that this value pertains to whole-brain segmentation only, not to individual anatomical labels:

"This Dice value refers specifically to the whole-brain extraction step, and not the individual anatomical labels" (lines 545-546).

On page 17 it is stated that: "TEDS-Net achieved an average Dice overlap across all structures of 0.78 ± 0.03 on the unseen test dataset, A full description of the network performance in shown in Suppl Fig. 7" Again this appears to be a summary average for some labels: it is not clear how they relate to the results measures.

Two networks were used in this study: a U-Net for whole-brain extraction (Dice = 0.96 ± 0.01 , 70 test scans) and TEDS-Net for cortical and subcortical segmentation. The value of 0.78 ± 0.03 refers specifically to TEDS-Net performance across all 10 anatomical regions included in this work. Per-structure Dice coefficients are provided in

Supplementary Fig. 9, and we have clarified in the Methods that the reported average corresponds to the mean Dice across these 10 regions:

“TEDS-Net was used to directly segment the 10 cortical and subcortical regions delineated on the atlases: CoP, WM, DGM, CB, Th, LV, ChP, FH, BS, and CSP. On the unseen D_{testing} , it achieved an average Dice overlap of 0.78 ± 0.03 across these structures, with per-structure scores provided in Suppl. Fig. 9. Visual inspection of the test dataset’s segmentation masks further confirmed that no manual refinement was necessary.” (lines 564-69).

Supplementary figure 7 shows DICE values for some whole brain regions, but should be changed to show the DICE values for each of the volumetric regions presented in the results (at least all the volume measures by lobe mentioned listed in Table 1). This figure should also indicate clearly the full range of the DICE values (it is not clear if the bars indicate standard deviation or range or what the other dots indicate).

The low average values of DICE (Supplementary Fig 7) for relatively large regions such as the whole cortical plate (0.6 at 18GW) is a real concern. Information about the sub regions actually used (eg lobe) estimates that are presented in the main results (and their max and min) is clearly needed here as they may well be even worse.

Note that, due to additional supplementary figures, Suppl. Fig 7 is now Suppl. Fig. 9.

As discussed above, all available Dice measurements are reported in Supplementary Fig 9. Regarding the cortical sub-regions (lobes), we did not create separate manual ground truth labels. Instead, the lobar parcellations were generated by warping the atlas-based parcellation templates to each individual scan, following the standard atlas-propagation approach widely used in MRI neuroimaging. As such, per-lobe Dice scores cannot be directly reported, as there is no independent manual reference for those regions.

However, we acknowledge there is some confusion here. To clarify this, we have revised the Methods to state explicitly that lobar parcellations were derived from atlas propagation rather than independently hand-labelled ground truth, and that Dice values reported in Supplementary Fig. 9 refer only to the primary anatomical segmentations:

“Lobar parcellations (frontal, temporal, parietal, occipital, insular) were generated by warping the 22 weeks’ gestation atlas-based parcellation templates to each individual scan using the continuous diffeomorphic deformation learnt in TEDS-Net. As these lobar labels were derived from propagated templates rather than independently hand-delineated ground truth, per-lobe Dice coefficients are not reported. Dice metrics shown in Suppl. Fig. 7 therefore refer only to the manually delineated anatomical structures (lines 570-75)

In addition, we have clarified the boxplot representation in the figure caption:

“The boxplot displays the distribution of values: the box spans the interquartile range (IQR) between the first (Q1) and third quartiles (Q3), the line inside the box marks the median, whiskers extend to the most extreme data points within $1.5 \times \text{IQR}$, and any points beyond the whiskers are shown individually as outliers. Lobar parcellations (frontal, temporal, parietal, occipital, insular) were generated by diffeomorphic atlas-

based template propagation and are therefore not included in these Dice metrics.”
Caption Suppl. Fig 9

We agree that the Dice for the cortical plate is lower than for some other regions, particularly at earlier gestational ages, reflecting the challenges of delineating this fetal structure with ultrasound. However, we also note that Dice performance is strongly age-dependent, improving at later gestation ages when the cortical plate is more clearly defined.

Atlas Data Balance:

An important missing factor for the atlas/training subjects is what sub-groups/cohorts they were taken from: do the atlas labels used to train the segmentation allow an representative/unbiased segmentation of the potentially different cohorts? (full demographic information relating to the training and testing data should be provided: age/sex/race geographic area etc is needed and how it relates to the same demographics across full study data). This is particularly important if the authors intend the system to be used in other researchers that may have different cohort balances.

Given only one hemisphere is being measured the atlas data should also include hemispheric balance in relation to main demographic factors such as age/sex/race etc.

We thank the Reviewer for raising this point. Both the atlas and labelled training and testing datasets were derived from the same INTERGROWTH-21st FGLS cohort and, in our *Nature* paper (Supplementary Tables 3–5), we showed that the atlas sample was demographically representative of the full cohort in terms of maternal characteristics, fetal sex and study site. Hemispheric balance of the image-derived phenotypes is also documented in the present manuscript (see Supplementary Table 7), given that there is no systematic bias between left and right visible hemispheres with respect to age, sex or study site.

As the present study uses the same dataset and QC pipeline as the atlas paper, the representativeness and hemispheric balance demonstrated previously apply here as well. For clarity, however, we have now cited the relevant tables from the atlas paper in the revised manuscript.

“The atlases were derived from a subset of the INTERGROWTH-21st FGLS cohort, which was found to be demographically representative of the full FGLS cohort in terms of maternal characteristics, fetal sex and study site (see Supplementary Tables 3–5 in 1).”
(lines 506-9).

Further, to further show the balance in our training and testing datasets, we have now compiled demographic and cohort information for bot, including age, sex, study site, and visible hemisphere. This allows assessment of how representative the atlas-labelled subset is relative to the full study cohort. We have included the table below in the revised manuscript (Supplementary Table 10), enabling readers to evaluate cohort balance and potential biases.

Results

Atlas image comparison to MRI atlases: This is really useful!

We thank the Reviewer for their positive comment.

It is stated that an age range of 18-27 (abstract) weeks was analyzed in the main cohort, but the comparison of the ultrasound atlas used for with MRI results (Figure 4) apparently only includes 18-26 gestational weeks. This needs to be explained and/or corrected.

We thank the Reviewer for raising this concern. In this study, we included subject from 18^{+0days} until 26^{+6days} weeks' gestation, as written in Fig. 1, which is commonly written as 18 to 27 weeks' gestation (e.g., up to the 27th gestational week). The ultrasound and MRI atlases present the fetal brain across a weeks' period (e.g., a 1-week atlas spans X^{+0days} to X^{+6days}). Accordingly, the 24 week atlases shown in Figure 4 includes data up to 24^{+6 days}.

To clarify this, we have amended the caption to read:

"The volume measures of each atlas-labelled structure across the atlases. Each atlas spans a one-week period, e.g., the 26th-week atlas spans 26^{+0days} to 26^{+6days}." Caption Fig. 4

Presentation:

Page 16 states "As such, TEDS-Net yielded segmentation masks only for the visible hemisphere (distal to the US probe)." It is a concern that all the data presented is apparently only for one hemisphere and pooled (even given the arguments presented to support this). The figures of labels and results (eg figure 2) give the impression the whole brain was analyzed. The figures should be corrected to show the actual labelling scheme/region used for measurements.

We thank the Reviewer for recognising why it was only possible to analyse the hemisphere distal to the ultrasound probe because of acoustic shadows and reverberation artifacts. Given these technical issues, which are well described in the Methods, we feel that we have addressed this point and already provided sufficient justification for pooling size measures in the Discussion. Nevertheless, we have added to the text to strengthen the point:

"We pooled the size measures between hemispheres rather than report values for each hemisphere. However, this was justified as the standardised differences between the hemispheres were within the WHO recommended range for comparing human growth patterns (± 0.5 units of SD) and confirmed by consistent growth trajectories across both hemispheres (Suppl. Fig. 7). Importantly, this strategy is supported by our previous findings showing that the asymmetries in fetal brain structures during the gestational age window studied here are driven by shape, not size" (lines 390-7).

However, we acknowledge the Reviewer's concern that the figures of labels and results may give the impression that the whole brain was analysed. We have, therefore, revised Table 1 to indicate explicitly that measures were derived from the visible hemisphere only and updated the caption across the manuscript.

“TB and CSP was labelled across the entire brain, whereas, the remaining regions were only segmented on the distal hemisphere.” Caption Table. 1

11 regions of interest			5 cortical parcellations		
Both	TB	Total Brain	Distal hemisphere	FL	Frontal Lobe
	CSP	Cavum septum		PL	Parietal lobe
Distal hemisphere only	CoP	Cortical plate		OL	Occipital lobe
	WM	White matter		TL	Temporal lobe
	DGM	Deep grey matter		IL	Insular lobe
	CB	Cerebellum		Measurement type	
	ChP	Choroid plexus		V	Volume
	LV	Lateral posterior ventricle horns		SA	Surface area
	FH	Frontal horns		D	Depth
	BS	Brain stem		T	Thickness
	Th	Thalamus		SFD	Sylvian Fissure depth

“It should be noted that, although both the left and right hemispheres were labelled on each atlas, only the distal hemisphere was labelled on individual scans, with the exception of TB and CSP.” Caption Fig 2

It also needs to be explained how the hemisphere was chosen (particularly for images where there is some oblique angulation of the brain with the sensor).

To ensure that the correct hemisphere was consistently identified, we trained a deep learning classifier to distinguish left from right hemisphere. This was a straightforward task, as acquisition protocols were standardised in the INTERGROWTH-21st study, and the model achieved 99.9% accuracy on a held-out validation set. The classifier was then applied to the full dataset, and the resulting hemisphere labels were passed to the segmentation model to guarantee correct labelling, even in cases of oblique angulation. We have clarified this procedure in the Methods:

“As such, TEDS-Net yielded segmentation masks only for the visible hemisphere (distal to the US probe), shown in Table 1. To achieve this, distal hemisphere was used as an input into TEDS-Net, to ensure the correct side was labelled. Hemisphere labelling was performed using a deep learning classifier which achieved an accuracy of 99.9% on the held-out test set. In our dataset, more right hemispheres were imaged at sufficient quality than left (Supplementary Table X), consistent with previous reports that fetuses more often adopt a leftward head position at this gestational age range.” (lines 585-92)

It is argued that the hemispheric differences were small enough to ignore and data could be directly pooled. Given the sometimes low performance of the segmentation on only 70 cases, and the high variance and outliers seen in many of the results graphs, it would seem that stronger evidence is needed to support this decision for each of the measures over the entire developmental range studied. Given the amount of data available it would seem reasonable to include (perhaps in the supplementary sections) separate left and right hemisphere growth data for each of the label measures used in the study. To help support the accuracy of the measures of each region, the paper should provide the growth model trajectory fits (with confidence intervals) for both hemispheres of each label (not just percentiles) to confirm the left right consistency over the age range. This would help support

the claimed accuracy of all the results and allow comparison to earlier fetal brain studies.

We agree that separate left–right analyses are of interest. However, as noted in the Methods, only the hemisphere distal to the ultrasound probe can typically be imaged with sufficient quality, and the dataset is therefore imbalanced, with more right hemispheres available than left. This asymmetry likely reflects a fetal head orientation bias: longitudinal ultrasound studies have shown that fetuses more often adopt a rightward head position, which explains the imbalance in usable hemisphere data.

To address the Reviewer’s concern, we conducted additional analyses on the cortical plate, re-fitting growth models separately for left and right hemispheres (see Figures below). The resulting size trajectories were almost identical, with variation in cortical plate measures within each hemisphere substantially greater than any systematic difference between them. This is consistent with our previous report ², which demonstrated that fetal brain asymmetries in this gestational window are primarily driven by shape rather than size.

We have included these hemisphere-specific trajectories as a Supplementary figure 7, which further demonstrates that pooling across hemispheres is justified for the present analyses. This is now referenced in the discussion:

“However, this was justified as the standardised differences between the hemispheres were within the WHO recommended range for comparing human growth patterns (± 0.5 units of SD) and confirmed by consistent growth trajectories across both hemispheres (Suppl. Fig. 7).” (lines 391-95).

All figure captions of results should clearly state estimates are based only one hemisphere measurement where appropriate and whether the measure is doubled to match a full brain (volume) estimate or is just one hemisphere.

We have revised Table 1 to indicate explicitly that measures were derived from the visible hemisphere only. Further, we have updated the caption for each required figure.

“The volume measures of each atlas-labelled structure across both hemispheres for each atlas” Fig. 4

“All IDPs except TBV and CSPV, measure only from the distal hemisphere.” Fig. 6

“Each of these IDPs is measured from the distal hemisphere.” Fig. 7

“(a) Relative parietal lobe volume (rPLV) and relative insular lobe volume (rILV) from the distal hemisphere.” Fig. 8

Fig 7 has no numbers indicating gestational weeks marked on the horizontal axis.

We thank the reviewer for spotting this oversight, which has now been corrected.

Variability in final measures:

Given the deep learning methods used for automated labeling do not provide an indication of certainty in the labelling results and there was apparently an image QC procedure used for the entire dataset, the results do exhibit a high variance and apparently have some very extreme outliers (eg supplementary figure 4 and 5). An example is the basic measures of white matter volume normalized by total brain volume (which as large regions, so should be reasonably reliable) which appear to contain a number of cases where the ratio WM/TBV differs by a factor of 3 or more from the mean. It is therefore unclear how reliable the labeling and quality control are for the label results (particularly the smaller more challenging regions).

We thank the Reviewer for this observation. We believe much of the concern regarding apparent extreme variability arises from how the results were originally visualised. Previously, all 4,205 individual volume measurements were plotted as fully opaque scatter points, which obscured the density of the central distribution and made the few true outliers appear more prominent. We have regenerated these plots using 20% opacity, which more clearly illustrates that the bulk of the data are tightly distributed and that extreme outliers are rare (see Figures below).

In addition, as noted in the Methods, values lying more than 4 SD from the mean were excluded from growth modelling to minimise the influence of spurious measures. This conservative exclusion procedure ensures that a small number of outlier values (likely arising from either image quality limitations or atypical anatomy) do not bias the main results. The revised visualisation and clarification of the exclusion criteria together support the reliability of the labelling for the vast majority of the dataset, while still transparently acknowledging the presence of a few extreme values, most likely rare outliers.

Although the atlas volumes are compared to the MRI atlas volumes (eg figure 4), there is less discussion of how the actual results of the full study data relate to previously published MRI atlases for this age range. This is particularly important for measures that appear to vary significantly from previous MRI studies (for example cortical plate thickness).

We thank the Reviewer for this important suggestion. In the Discussion, we already compare several structures to previous MRI studies, but have clarified and added to the text. In addition, we reference our recent validation study in which we applied this segmentation pipeline alongside a leading MRI pipeline⁴ to same-day fetal MRI and ultrasound scans. In that work, image-derived phenotypes were highly correlated, with excellent-to-good ICC for subcortical structures. This strong correspondence supports the reliability of our pipeline and provides independent evidence that the recorded volumes are robust.

“For example, the median CoPV range of 3 to 18 cm³ reported here (Fig. 7) is comparable to the 5 to 22 cm³ range in a previous MRI study, with lobar volumes also showing high compatibility. Similar agreement was also seen for CBV and DGMV. Further, this approach was recently validated on an independent dataset of 90 fetal brain ultrasound volumes with paired, same-day MRI³, where volume measures were highly correlated, with good to excellent agreement in the subcortical structures.” (lines 333-8).

We agree, that further discussion would be valuable, particularly for measures such as cortical plate thickness where differences are more pronounced. Hence, we now explicitly compare, in the Discussion, our cortical plate thickness trajectories to published MRI studies, highlighting both methodological differences (e.g., ultrasound vs MRI resolution and contrast) and biological consistency where ranges overlap:

“We observed median cortical plate thickness values of 2-3 mm between 18 and 27 weeks, which is consistent with 1.92-2.84 mm reported in an MRI-based study⁵. However, cortical thickness estimates are variable across studies⁵⁻⁷. Differences in absolute values likely reflect modality- and method-related factors: in ultrasound, cortical plate boundaries are defined by echogenicity, whereas in MRI they are delineated using tissue relaxation contrasts. Such cross-modality variation has also been described for other fetal brain structures. We note, however, that cortical thickness estimates show greater variability than other measures, consistent with previous MRI reports, reflecting both methodological differences and the intrinsic difficulty of delineating this boundary during early development. Importantly, the developmental trajectories across gestation are consistent with MRI findings⁵, supporting the biological validity of our measures.” (lines 339-50)

The cortical thickness results are a particular concern as they indicate a very large variability in the average estimates and appear to indicate there is no significant increase in cortical thickness over the age range (and even a reduction at early time points) This is not mentioned or explained, and the results are perhaps not well supported by previous studies. The thickness of the CP appears large for this age range (between 2 and 2.5mm or greater even at 19 or 20 weeks gestation).

Cortical thickness measurements in the fetal brain are indeed challenging, and inconsistencies are also evident across MRI studies. For example, Corbett et al. (2011) reported median thickness values ranging from 1.92 to 2.84 mm between 20–26 weeks' gestation, similar to the range we observe. In contrast, Xu et al. (2022), studying a wider gestational age window (22–38 weeks), reported a much narrower range of 1.4–1.5 mm with large variability between subject. Our estimates are, therefore, comparable to some MRI studies but higher than others, highlighting methodological heterogeneity across modalities and cohorts.

Our approach to measuring cortical thickness also differs from MRI-based methods. As ultrasound imaging typically captures only one hemisphere at sufficient quality, we cannot apply surface-based potential field models commonly used in MRI (e.g., Jones 2000). Instead, we derived cortical thickness directly from voxel-wise distance measures, which may partly explain any discrepancies with MRI-derived values.

Importantly, in our recent cross-modality validation, cortical plate volumes were highly correlated with MRI but systematically larger, consistent with the widely recognised observation that the cortical plate appears thicker on ultrasound compared to MRI. We have explained in the text that these methodological and modality-related factors likely contribute to the variability in thickness estimates, while the overall developmental trajectories remain biologically plausible, as expressed in the text we have added to the Discussion:

“We observed median cortical plate thickness values of 2-3 mm between 18 and 27 weeks, which overlap with 1.92-2.84 mm reported in an MRI-based study. However, cortical thickness estimates are variable across studies. Differences in absolute values likely reflect modality- and method-related factors: in ultrasound, cortical plate boundaries are defined by echogenicity, whereas in MRI they are delineated using tissue relaxation contrasts. Such cross-modality variation has also been described for other fetal brain structures⁵⁵. We note, however, that cortical thickness estimates show greater variability than other measures, consistent with previous MRI reports, reflecting both methodological differences and the intrinsic difficulty of delineating this boundary during early development. Importantly, the developmental trajectories across gestation are consistent with MRI findings¹⁹, supporting the biological validity of our measures.” (lines 339-50)

There is no clear description of how CP thickness was calculated or validated so it is difficult to see how these results relate to previous findings.

We have now expanded the Methods to provide a clearer description of how cortical plate thickness (CoPT) was calculated:

“CoPT was measured at each point along the cortical surface by computing the orthogonal projection from the surface to the nearest boundary point along the gradient vector, and averaging across the cortical plate. This voxel-wise distance transform approach achieves the same goal as MRI-based cortical thickness estimation but uses a different implementation since surface-based potential field models commonly applied in MRI cannot be used reliably in ultrasound. To validate these measures, we confirmed that CoPT estimates were within the 1.5-3.0 mm range reported in MRI atlases for 18-27 weeks’ gestation^{5,7} and showed consistent age-related trends across the cohort” (lines 613-20).

Overall, given the large number of measures evaluated, it would perhaps be useful to focus the paper on those measures that are most accurately/precisely estimated from ultrasound and to not include those (for example cortical thickness) that are perhaps more questionable when estimated from ultrasound studies.

We thank the Reviewer for this suggestion. We agree that some measures, such as cortical thickness, are more challenging to estimate from ultrasound than others, and we have therefore clarified these limitations in the revised Methods and Discussion. Nonetheless, we believe it is important to report all measures produced to avoid reporting bias. As far as cortical thickness is concerned, we consider it very important as our study represents one of the first large-scale attempts to quantify this measure using ultrasound. Our estimates fall within the range reported in MRI studies for the same gestational window, and the developmental trajectories are biologically plausible, though we acknowledge the variability is higher than for other measures. Including these data provides a valuable reference point for future studies and enables direct comparison with MRI-derived phenotypes.

Reviewer #1 (Remarks on code availability):

I briefly looked at the github pages. They contain instructions and test data to support use of the code. The installation of the code was relatively complex, requiring additional libraries so I could not run and verify the code myself on my home laptop. Given time and expertise in python library versions it seems feasible to get this running using the documentation provided.

We thank the Reviewer for their feedback on the GitHub repository. We acknowledge that installation currently requires several dependencies, and we will improve the documentation and streamline installation prior to publication to make the code easier to use before release.

Reviewer #2 (Remarks to the Author):

While the study offers several strengths—including a large, multi-centre cohort, state-of-the-art deep-learning segmentation, and the first normative 3D-ultrasound growth curves for 16 fetal brain structures—two limitations temper enthusiasm.

We thank the Reviewer for their positive comments and we address the perceived limitations individually below.

1) The manuscript presents only "prescriptive" trajectories derived from low-risk pregnancies, omitting a comparison arm of fetuses with known neuro-developmental abnormalities. Given that abnormal brain development may follow divergent volumetric paths, the absence of such a contrast group precludes assessment of whether the reported percentiles retain discriminative validity at the pathological end of the spectrum. Consequently, the clinical incremental value over their 2023 study is limited.

We thank the Reviewer for their interesting comments. Our 2023 *Nature* paper presented a *qualitative* digital atlas of the normative spatiotemporal dynamics of fetal brain maturation. As such, the atlas has limited clinical value but may “*be useful as a research tool to investigate the fetal origins of neurodevelopmental disorders*” as stated at the end of the paper ². The present manuscript, on the other hand, describes *quantitatively* the growth of 16 key fetal brain structures associated with satisfactory domain-specific neurodevelopmental scores at 2 years of age. Although the region-specific, normative growth trajectories, described in the present manuscript, may provide a more refined research tool than the atlas alone, their potential as a clinical tool may be far more important.

We would argue that stating these are “*only “prescriptive” trajectories*” underestimates the enormous conceptual leap that the INTERGROWTH-21st Project represents in biomedicine and early child development. There were not such “*only prescriptive*” trajectories pertaining to fetal growth with normative developmental outcomes at 2 years of age until we introduced the concept in 2014 ⁸⁻¹¹, based on WHO recommendations for the construction of international, prescriptive, growth standards.

We agree with the Reviewer that the trajectories presented here will need to be evaluated in a more general population with higher and diverse developmental outcomes. Fortunately, the INTERBIO-21st Fetal Study ¹² gives us access to such a dataset and we are already conducting exploratory analyses. That work, which has different objectives, would add considerable length to the present manuscript. For that reason, we shall be submitting the results in a separate manuscript.

2) The authors excluded 101 infants with 2-year INTER-NDA scores below the 3rd percentile to preserve the atlas’s “normative” nature. However, the intracranial growth trajectories of these excluded infants should have been explicitly plotted against the retained cohort to quantify their deviation from normal centiles. If trajectories of the excluded cohort do not diverge significantly, the current imaging-derived phenotypes (e.g., rILV:rPLV) may lack sensitivity for early pathology; conversely, marked deviations would substantiate their utility as early biomarkers. This omission represents a missed opportunity for clinical validation.

The Reviewer makes an interesting point regarding the evaluation of a screening tool; however, we do not believe that the approach suggested would be appropriate: 1) there are many risk factors and aetiological pathways affecting developmental scores at 2 years of age. Hence, requiring this ratio to show deviation in such a heterogeneous group would, in our opinion, be an unfair requirement for the test; and 2) developmental scores at 2 years of age are also related to factors other than early brain growth velocity, such as exposures in postnatal life.

Therefore, before evaluating the ratio's predictive capacity, a detailed analysis of its distribution in the general population is required, as well as an understanding of the associated outcomes and postnatal timing of the appearance of any effect in a different population.

Of course, as stated in response to Reviewer 3's comment 2, we fully agree with the need to evaluate the predictive capacity of the tools we have presented; however, we believe this analysis cannot be conducted using the small sub-sample of infants excluded from this cohort because of low developmental scores at 2 years of age.

Nevertheless, to address the Reviewer's understandable concerns about the treatment of the cases in the present study, we have performed a sensitivity analysis in which the 101 excluded infants were included in the normative cohort, i.e., a kind of intention-to-treat approach. The growth trajectories were very similar to those obtained when these cases were excluded, with no evidence of systematic divergence. This demonstrates that the findings at hand are robust to the inclusion or exclusion of these cases. Formal evaluation of predictive performance will require larger, dedicated clinical studies.

Additional points:

1) The discussion claims that “observed differences ... are primarily due to socioeconomic, educational and class disparities, rather than genetic variants.”

This sweeping statement requires formal mediation analysis before it can be asserted with confidence.

We agree with the Reviewer that the reference cited does not adequately substantiate the statement. To support the concept that socioeconomic, educational and class disparities have a greater influence on development than genetic ancestry, we have instead cited the Noble et al. *Nature Neuroscience* (2015;18,773-8) paper and the seminal *PNAS* paper of the Nobel Laureate, James Heckman (2007;104:13250-5). We have also used the term “*genetic ancestry*” rather than “*genetic variants*”.

“Finally, our findings reinforce a fundamental biological principle: most of the observed differences in growth and neurodevelopment across populations or regions of the world are primarily due to socioeconomic, educational and class disparities, rather than genetic ancestry¹³⁻¹⁵. The notion that brain size and neurodevelopment are related to skin colour or ‘race’ is simply false.” (line 421-5)

2) The statistical section would benefit from review by an independent biostatistician, particularly to confirm the adequacy of the fractional-polynomial assumptions and the handling of repeated-measures correlation in the growth models.

We would like to reassure the Reviewer that our four statisticians are leading experts in their fields with well-established reputations. They are Eric Ohuma (Professor of Medical Statistics & Epidemiology, London School of Hygiene & Tropical Medicine) and Robert Gunier (Associate Director CERCH, School of Public Health, UC Berkeley) who have been our principal biostatisticians for many years. Their methods, used throughout this manuscript, are robust, have been applied across multiple growth measures, and have been peer-reviewed in *Lancet* and *Nature* journals. For the neuroimaging analyses described in the present manuscript, we are fortunate to have received a significant contribution from Thomas Nichols (Professor of Neuroimaging Statistics, Big Data Institute, University of Oxford) and Stephen Smith (Professor of Biomedical Engineering, Wellcome Centre for Integrative Neuroimaging, University of Oxford). Professor Nicholls was formerly the Director of Modelling and Genetics at GSK’s Clinical Imaging Centre. In Oxford, he is developing advanced statistical and machine learning methods for brain image data. Professor Smith is the head of the Analysis Group at the Wellcome Centre for Integrative Neuroimaging and developed the FSL Package, with complex statistical modelling at its core. It used by 1,000 labs worldwide (140,000 citations).

Reviewer #3 (Remarks to the Author):

This is a follow-up analysis of a previously published large-scale, multinational study based on fetal brain 3D ultrasound scans and advanced reconstruction methods. The authors build normative growth trajectories, introduce a novel “fetal brain maturation index” and propose the rILV:rPLV ratio as a potential functional correlate. The work is validated against neurodevelopmental outcomes at 2 years of age. There are some methodological and theoretical considerations that merit further attention and clarification:

We thank the Reviewer for their positive comments and address the methodological and theoretical considerations individually below.

1. The authors evaluated site effects using two strategies:

(1) a leave-one-site-out approach: it is unclear how the authors quantitatively assessed the statistical significance of excluding each site. Could they provide the growth curves for each site individually to better visualize local deviations?

We apologise if this was unclear. Figure 5a already presents the leave-one-site-out growth curves for four sites, illustrating the effect of excluding individual sites on the centile estimates. As shown, the trajectories are highly consistent across sites, with minimal local deviations, supporting the pooling of data.

(2) the SSD (standardized site difference) metric: the metric merges within gestational windows, thereby ignoring potential site-by-age interactions (i.e., site effects on the developmental trajectory rather than a specific time point). It is also discussed by the authors that inconsistencies in mean gestational age disturbed the comparisons (line 169). Do the authors consider applying more modern nonlinear harmonization frameworks such as ComBat-GAM to more robustly address site and scanner heterogeneity.

We thank the Reviewer for raising this issue. In an earlier publication⁸, we adjusted fetal growth measures to the midpoint of each gestational age window to reduce bias from site differences in mean gestational age. The relevant sentence from that publication was:

“We adjusted the fetal head circumference measures at each site to the midpoint of each gestational age interval by use of estimates obtained from fitting a fractional polynomial regression model, assuming the growth rate was uniform within each of the 5 week intervals”.

In the present study, we used GAMLSS modelling, which explicitly accounts for non-linear age-related changes in both the mean and variance of the data. This framework already addresses the issue of site-by-age interactions when modelling developmental trajectories.

We acknowledge that ComBat-GAM has been successfully applied to large MRI datasets to harmonise site and scanner effects. However, our variance component analysis showed that study site explained only 0.6–5.8% of the total variance in brain measures, indicating that any residual site effect was minimal. We therefore judged that additional harmonisation would be unlikely to affect the results materially. For clarity, we have now added text to the Discussion to explain why GAMLSS was chosen and why ComBat-GAM was not applied:

“In addition, our use of GAMLSS explicitly models non-linear developmental trajectories and variance across gestation, thereby mitigating potential site-by-age confounding effects without the need for additional harmonization methods” (lines 301-4).

In addition, the authors included site effect in the subsequent mixed-effects model. Were site effects statistically significant in these models?

Site was included as a random effect in the mixed-effects models, and its contribution to variance was quantified through variance component analysis (VCA), as reported in Supplementary Table 4. Across the 28 IDPs, the proportion of variance explained by site was small (1.6% for TBV and <5.8% for all other structures; Fig. 5b). In most cases, site effects were not statistically significant once sex and gestational age were included, consistent with the minimal variance explained. We have clarified this point in the Results:

“Consistent with this minimal variance explained, site effects were not statistically significant in the mixed-effects models once sex and gestational age were included” (lines 183-5).

2. While the study used a prescriptive, “healthy” cohort to build normative models, some individuals in the broader cohort had low neurodevelopmental scores were excluded from model construction. It would greatly strengthen the manuscript to test the brain maturation index on these children with suboptimal neurodevelopmental outcomes,

We fully agree with the need to evaluate the predictive capacity of the tools we have presented, which would primarily be an evaluation of cut-off points in relation to specific outcomes. However, we believe this analysis cannot be conducted using the small subsample of infants excluded from this cohort because of developmental scores at 2 years of age, as there are many risk factors and aetiological pathways affecting developmental scores at 2 years of age and developmental scores at 2 years of age are also related to factors other than early brain growth velocity, such as exposures in postnatal life. Nevertheless, to address the Reviewer’s point, we have revised the relevant sentence in the Discussion to:

“We plan to use the index in the future to quantify the degree of departure from chronological age across the participants in the INTERBIO-21st Fetal Study³⁷ who have different risk profiles at this key stage of brain development” (lines 295-7).

or on individuals with diagnosed neurodevelopmental disorders, to evaluate the model’s generalizability and potential for early risk screening.

We fully agree with the Reviewer: this is an essential step before clinical implementation, as we also noted in our response to Reviewer 2 (comment 1). However, the analysis requires a completely different population, e.g. a cohort from the general population with a wide range of developmental outcomes. The INTERBIO-21st Fetal Study¹² gives us access to such a dataset and we are already conducting exploratory analyses. That work, which has different objectives, would add considerable length to the present manuscript. For that reason, we shall be submitting the results in a separate manuscript.

3. The authors presented the 3rd and 97th centiles as normative boundaries. However, certain metrics, for example FLT, PLT, and OLT in Fig. 7, show

distributions that appear non-Gaussian at younger gestational ages. The 97th centile curve in particular seems to deviate far from the observed scatter, suggesting a potential misfit of the normal-based percentile approach. I suggest the authors to more formally examine distributional normality across gestational ages, and to consider alternative methods that would not assume Gaussianity, in order to derive more accurate and clinically meaningful centile boundaries.

We agree that some measures, particularly cortical thickness metrics at earlier gestational ages, show distributions that deviate from normality. To account for this, we did not rely on simple Gaussian assumptions but instead modelled centiles using flexible approaches that allow for skewness and kurtosis. In particular, we used the GAMLSS framework, which estimates smooth gestational-age-dependent parameters for the mean, variance, skewness, and kurtosis of each distribution. This approach has been widely applied in the production of growth standards, including previous INTERGROWTH-21st charts, and provides robust centile estimation even when the underlying distribution is non-Gaussian. We have clarified this in the Statistical Analysis section of the Methods:

“To account for the increasing inter-subject variability with gestational age, the mean and standard deviation (SD) were modelled separately using fractional polynomial regression within the GAMLSS framework in R. This approach also allows for modelling of higher-order distributional parameters, such as skewness and kurtosis, thereby accommodating non-Gaussian distributions and providing robust estimations of centiles” (lines 687-94).

4. Although the authors collected data up to 30 weeks’ gestation, the modeling focused only on 18–27 weeks. The rationale for selecting this window is well argued, but the exclusion of later scans might limit applicability for clinical follow-up into the third trimester. If data quality permits, the authors could consider extending their models toward the late second and third trimester in future work to provide more comprehensive reference trajectories.

We thank the Reviewer for this constructive suggestion. We agree that extending trajectories into the late second and third trimester would provide valuable reference data. Our present analyses already provide comprehensive coverage of the second trimester, a critical developmental window during which faltering cranial growth has been shown to predict later neurodevelopmental outcomes¹². The main limitation beyond this period is image quality: increasing calcification of the fetal skull bones in the third trimester produces acoustic shadowing and reverberation artefacts that severely restrict the reliable visualisation of intracranial structures. For this reason, we focused on the 18–27 week window, which yields consistently high-quality data. We agree that if future methodological advances allow reliable imaging later in pregnancy, extending these normative trajectories into the third trimester will be an important next step.

5. The segmentation procedure was performed only on the distal hemisphere due to acoustic shadowing constraints. While a “distal hemisphere” confounder was included in the regression models, the study does not explicitly analyze potential lateralization or asymmetry effects, which could be biologically meaningful in early fetal cortical development.

We agree that potential hemispheric asymmetries could be biologically meaningful. However, the direction of the fetal head (left or right distal hemisphere) is determined by posture and probe orientation, and there is no evidence that it is related to underlying

measures of brain development. To verify this, we compared growth trajectories of the cortical plate between left and right hemispheres in our dataset (Supplementary Figure 7). The resulting trajectories were highly similar, with variation within each hemisphere substantially greater than any systematic left–right differences. This is consistent with our previous report ², which demonstrated that during this gestational age window, hemispheric asymmetries are primarily driven by shape rather than size. These findings support our assumption that the choice of distal hemisphere does not bias the reported developmental measures.

6. The proposed rILV:rPLV ratio shows promise as an indicator of asynchronous cortical maturation, but the manuscript stops short of demonstrating its relationship with functional outcomes. Could the authors consider an exploratory analysis correlating rILV:rPLV deviations with neurodevelopmental scales to support its potential clinical utility?

We fully agree with the Reviewer. The analysis suggested would be an essential step before clinical implementation; however, for the reasons given above in response to the Reviewer’s comment 2, we do not believe it is practical to include such an analysis in the present manuscript.

References

- 1 Moser, F., Huang, R., Consortium, I.-., Papiez, B. W. & Namburete, A. I. L. BEAN: Brain Extraction and Alignment Network for 3D Fetal Neurosonography. *Neuroimage* **258**, 119341 (2022). <https://doi.org/10.1016/j.neuroimage.2022.119341>
- 2 Namburete, A. I. L. *et al.* Normative spatiotemporal fetal brain maturation with satisfactory development at 2 years. *Nature* **623**, 106-114 (2023). <https://doi.org/10.1038/s41586-023-06630-3>
- 3 Wyburd, M. K. D., Nicola K.; Kyriakopoulou, Vanessa; Venturini, Lorenzo; Wright, Robert; Uus, Alena; Matthew, Jacqueline; Skelton, Emily; Zöllei, Lilla; Hajnal, Joseph; Namburete, Ana I. L. Cross-Modality Comparison of Fetal Brain Phenotypes: Insights from Short-Interval Second-Trimester MRI and Ultrasound Imaging. *Human Brain Mapping - In Press* (2025).
- 4 Uus, A. U. *et al.* BOUNTI: Brain vOLumetry and aUtomated parcellation for 3D fetal MRI. *bioRxiv* (2023). <https://doi.org/10.1101/2023.04.18.537347>
- 5 Corbett-Detig, J. *et al.* 3D global and regional patterns of human fetal subplate growth determined in utero. *Brain Structure and Function* **215**, 255-263 (2011).
- 6 Xu, F. *et al.* Morphological development trajectory and structural covariance network of the human fetal cortical plate during the early second trimester. *Cerebral Cortex* **31**, 4794-4807 (2021).
- 7 Xu, X. *et al.* Spatiotemporal atlas of the fetal brain depicts cortical developmental gradient. *Journal of Neuroscience* **42**, 9435-9449 (2022).

- 8 Villar, J. *et al.* The likeness of fetal growth and newborn size across non-isolated populations in the INTERGROWTH-21st Project: the Fetal Growth Longitudinal Study and Newborn Cross-Sectional Study. *Lancet Diabetes Endocrinol* **2**, 781-792 (2014). [https://doi.org/10.1016/S2213-8587\(14\)70121-4](https://doi.org/10.1016/S2213-8587(14)70121-4)
- 9 Villar, J. *et al.* International standards for newborn weight, length, and head circumference by gestational age and sex: the Newborn Cross-Sectional Study of the INTERGROWTH-21st Project. *Lancet* **384**, 857-868 (2014). [https://doi.org/10.1016/S0140-6736\(14\)60932-6](https://doi.org/10.1016/S0140-6736(14)60932-6)
- 10 Papageorgiou, A. T. *et al.* International standards for fetal growth based on serial ultrasound measurements: the Fetal Growth Longitudinal Study of the INTERGROWTH-21st Project. *Lancet* **384**, 869-879 (2014). [https://doi.org/10.1016/S0140-6736\(14\)61490-2](https://doi.org/10.1016/S0140-6736(14)61490-2)
- 11 Papageorgiou, A. T. *et al.* International standards for early fetal size and pregnancy dating based on ultrasound measurement of crown-rump length in the first trimester of pregnancy. *Ultrasound Obstet Gynecol* **44**, 641-648 (2014). <https://doi.org/10.1002/uog.13448>
- 12 Villar, J. *et al.* Fetal cranial growth trajectories are associated with growth and neurodevelopment at 2 years of age: INTERBIO-21st Fetal Study. *Nat Med* **27**, 647-652 (2021). <https://doi.org/10.1038/s41591-021-01280-2>
- 13 Heckman, J. J. The economics, technology, and neuroscience of human capability formation. *Proceedings of the national Academy of Sciences* **104**, 13250-13255 (2007).
- 14 Noble, K. G. *et al.* Family income, parental education and brain structure in children and adolescents. *Nature neuroscience* **18**, 773-778 (2015).
- 15 Graham, G. N. Why your ZIP code matters more than your genetic code: promoting healthy outcomes from mother to child. *Breastfeeding Medicine* **11**, 396-397 (2016).

"Normative growth trajectories of fetal brain regions validated by satisfactory maturation of neurodevelopmental domains at 2 years of age"

REVIEWER COMMENTS

Reviewer #1 (Remarks to the Author):

Overall, the authors have done well to clarify my misunderstanding and confusion and resolve the concerns I had. I have a couple of minor follow ups to the rebuttal that should be easy to address:

Contrast normalization:

With regard to the response to my question regarding intensity normalization (page 4), you could perhaps just make it simply clear in the response that. (I think that:?) unlike many standard MR image processing pipelines which often make use of a tissue intensity standardization (eg of CSF or white matter intensity) and/or a bias correction step, you make use of training across a large dataset with varying contrast, to deep-learn a contrast independent labeling and therefore negate the impact of potentially varying tissue/boundary contrast on the ultrasound segmentation process. This seems to be the case? (if you are just making the maximum and minimum in the entire volume to be 1 and 0)?

We thank the reviewer for this helpful clarification. The reviewer's understanding is correct.

Unlike MRI, where tissues intensities are strongly influenced by acquisition parameters (e.g., T1/T2 weighting, coil sensitivity, and bias-field homogeneities), ultrasound intensities are physically related to the acoustic impedance and reflection properties of underlying structures. Nevertheless, ultrasound image contrast can still vary due to acquisition settings (e.g., gain compensation, angle of insonation, and probe position). In the INTERGROWTH-21st FGLS dataset, the scanning protocol and gain calibration were standardised across all sites and gestational ages (Line 481 -7), and protocol adherence was regularly monitored by a central quality control team. This ensured that raw intensity distributions were relatively stable across the dataset. Any remaining variation was handled through per-volume min-max normalisation (scaling each image to the [0,1] range), and by training the deep learning model on a large, heterogeneous dataset. This combination allows the model to learn features that are robust to residual contrast variation, effectively achieving contrast invariance through exposure to naturally diverse examples during training.

To verify that min-max scaling preserved biologically meaningful contrast relationships, we examined the average intensities for representative tissue classes (cortical plate (CoP); white matter (WM); and Choroid Plexus (ChP)), before and after normalisation. We masked each volume in the training set using the ground-truth segmentation labels and measured the average intensity before and after min-max normalisation. The three tissues exhibited distinct, well-separated intensity distributions that retained their relative ordering after normalisation (Supplementary Fig. 8). These findings confirm that min-max scaling standardises image intensity while preserving the contrast structure required for

segmentation. Thus, additional tissue-based intensity normalisation or bias-field correction was unnecessary in this context.

To ensure clarity for readers who may be more familiar with MRI pipelines, we have now added the following statement to the manuscript and included the figure below to the Supplementary Materials:

Line 523-8 *“Tissue intensities were found to be relatively consistent across the dataset (see Supplementary Fig. 8), reflecting the standardised scanning protocol and the fact that ultrasound contrast arises from the physical acoustic properties of tissues. Therefore, more complex MRI-style intensity standardisation was not required; simple per-volume min–max normalisation was sufficient to standardise input scale while preserving relative contrast.”*

Tissue Intensities after min-max normalization

Image Resolution:

On page 3 of the rebuttal in response to my question: “How did the anisotropic resolution vary wrt anatomical axis in the different datasets?”

I did not quite understand the answer (perhaps I just need something simpler?):

Beyond stating that a standard protocol was used to collect the slices: It was not clear from the response and the paper whether that meant the collected slices (with a given thickness and in plane resolution) were always orientated precisely axially or did the anatomical orientation of the slices depend significantly on the fetal head orientation for each baby? i.e. Did the operator always manage to acquire approximately axial slices? Or was it very (or somewhat) baby + operator dependent? (do you have an approximate idea how big were the rotations away from precisely axial ?)

This is important because in your response you also mention that some datasets had a slice thickness down to 0.27mm while others had a slice thickness of up to 1.91mm. This is a considerable range (for example this is generally kept much more constrained in many MRI studies). As a result the slice orientation may impact the performance of the

segmentation.

We thank the reviewer for this clarification and now understand where the confusion may have arisen.

Unlike fetal MRI, where the fetus can adopt any arbitrary orientation within a fixed scanner, fetal ultrasound allows the operator to actively position both the probe (and, to some extent, the fetus) to acquire a specific anatomical plane. In this study, sonographers followed a standardised acquisition protocol in which the probe was positioned such that the central axial view was collected at the level of the thalami (Line 481 -7). As a result, although small variations are inevitable, the x -direction of the raw acquired volumes is generally well aligned with the axial plane of the brain. This active control over the acquisition plane is a key difference from fetal MRI.

To assess the degree of residual variability in practice, we examined the rotation parameters obtained during the alignment step in our preprocessing (Moser et al., 2020). These parameters describe the rotations required to align each scan to a common reference space. Across all 4,205 scans, the distributions for each of the three rotation angles showed two sharp, symmetric peaks separated by π , corresponding to the fetus facing left versus right. Importantly, the narrowness and consistency of these peaks indicate that the acquired volumes were oriented very similarly across the dataset, with only modest deviations (approximately within 10-15°) from the intended axial plane. It is also important to note that the “slice thickness” reported here refers to voxel spacing in the reconstructed 3D volume, not physical slice separation as in MRI. Therefore, variability in this dimension reflects differences in acquisition resolution rather than true angular misalignment or missing anatomical information.

Therefore, although voxel spacing varied across acquisitions, the anatomical orientation of the slices was highly consistent.

To make this clear for readers who are less familiar with ultrasound, we have added the following to the methods:

Line 500-11 *“The images were then resampled to an isotropic voxel size of $0.6 \times 0.6 \times 0.6 \text{ mm}^3$ using trilinear interpolation. Across the dataset, initial voxel sizes ranged from 0.14-0.76 mm (SD 0.04) in the x -dimension, 0.30-1.01 mm (SD 0.15) in the y -dimension, and 0.27-1.91 mm (SD 0.23) in the z -dimension, with a median of $0.32 \times 0.51 \times 0.85 \text{ mm}^3$. As the scans were collected at the axial view of the fetal head, the x -direction in the raw volumes consistently corresponds to the axial plane. This was confirmed empirically by examining the rotation parameters from the alignment step across all 4,205 scans, which showed two narrow peaks for separated by π (or 180°), corresponding to fetuses facing left versus right. The*

narrowness of these peaks indicates only modest deviation from the intended axial orientation (Supplementary Fig. 8). In 3D ultrasound, the reported “slice thickness” reflects voxel spacing in the reconstructed volume, not physical slice separation as in MRI, and therefore does not imply angular misalignment.”

Cortical thickness:

Your new description of the calculation of cortical thickness should perhaps include a reference to the method? I assume it has been used previously? It also appears that the thickness was across the discretely labelled CP voxels (did not account for any sub-voxel partial volume effects) and therefore will only be accurate to +/- a voxel (?) at any given point? (even if the automated labeling is perfect). If so, you should make this limitation clear in the description.

We thank the reviewer for this thoughtful suggestion.

Cortical thickness has not previously been measured from fetal ultrasound. As only a single hemisphere is typically visible, standard MRI surface-based methods could not be applied (as discussed in the Methods). To address this, we developed a voxel-wise approach inspired by established MRI methods that estimate thickness by integrating along a gradient vector between boundaries^{1,2}. We have now added the relevant references and clarified this in the Methods:

Line 652-55 “CoPT was measured at each point along the cortical surface by computing the orthogonal projection from the surface to the nearest boundary point along the local gradient vector, inspired by previous voxel-wise techniques for estimating CoPT in MRI^{77,78}.”

We agree with the reviewer that this method has inherent limitations. Voxel-wise thickness estimation is constrained by voxel size, segmentation accuracy, and partial-volume effects, as discussed in prior reviews³. We now explicitly acknowledge this limitation in the Discussion:

Line 370-4 “Because cortical thickness was computed voxel-wise from the CoP label, precision is inherently limited by voxel size (± 0.6 mm in this study), segmentation accuracy, and potential partial-volume effects³. In future work, adapting surface-based techniques to operate robustly on single-hemisphere ultrasound data may help address these limitations⁴.”

Reviewer #2 (Remarks to the Author):

Thank you for your reply, No more question.

Reviewer #3 (Remarks to the Author):

This is an insightful study offering large-scale, multi-centered fetal brain data with

comprehensive growth trajectories, providing a view of normative fetal brain development.

The authors have addressed my comments appropriately by providing supplementary statistics and by extensively discussing the possible limitations and future directions - the manuscript has been greatly improved.

We thank Reviewer #2 and #3 for revisiting our manuscript and these positive comments.

- 1 Miller, M. I., Massie, A. B., Ratnanather, J. T., Botteron, K. N. & Csernansky, J. G. Bayesian construction of geometrically based cortical thickness metrics. *NeuroImage* **12**, 676-687 (2000).
- 2 Hutton, C., De Vita, E., Ashburner, J., Deichmann, R. & Turner, R. Voxel-based cortical thickness measurements in MRI. *Neuroimage* **40**, 1701-1710 (2008).
- 3 Clarkson, M. J. *et al.* A comparison of voxel and surface based cortical thickness estimation methods. *Neuroimage* **57**, 856-865 (2011).
- 4 Jones, S. E., Buchbinder, B. R. & Aharon, I. Three-dimensional mapping of cortical thickness using Laplace's equation. *Human brain mapping* **11**, 12-32 (2000).